# Lion Secretly Solves Constrained Optimization, As Lyapunov Predicts

**Lizhang Chen**$^*$     **Bo Liu**$^*$     **Kaizhao Liang**$^*$     **Qiang Liu**
The University of Texas at Austin
{lzchen,bliu,kaizhaol,lqiang}@utexas.edu

## ABSTRACT

Lion (Evolved Sign Momentum), a new optimizer discovered through program search, has shown promising results in training large AI models. It performs comparably or favorably to AdamW but with greater memory efficiency. As we can expect from the results of a random search program, Lion incorporates elements from several existing algorithms, including signed momentum, decoupled weight decay, Polak, and Nesterov momentum, but does not fit into any existing category of theoretically grounded optimizers. Thus, even though Lion appears to perform well as a general-purpose optimizer for a wide range of tasks, its theoretical basis remains uncertain. This lack of theoretical clarity limits opportunities to further enhance and expand Lion's efficacy.

This work aims to demystify Lion. Based on both continuous-time and discrete-time analysis, we demonstrate that Lion is a theoretically novel and principled approach for minimizing a general loss function $f(x)$ while enforcing a bound constraint $\|x\|_\infty \leq 1/\lambda$. Lion achieves this through the incorporation of decoupled weight decay, where $\lambda$ represents the weight decay coefficient. Our analysis is made possible by the development of a new Lyapunov function for the Lion updates. It applies to a broader family of Lion-$\mathcal{K}$ algorithms, where the $\text{sign}(\cdot)$ operator in Lion is replaced by the subgradient of a convex function $\mathcal{K}$, leading to the solution of a general composite optimization problem of $\min_x f(x) + \mathcal{K}^*(x)$. Our findings provide valuable insights into the dynamics of Lion and pave the way for further improvements and extensions of Lion-related algorithms.

## 1 INTRODUCTION

Optimization serves as the cornerstone in training contemporary AI models. Given the immense computational demands associated with training large AI models, the design of an effective optimizer emerges as a paramount endeavor.

Traditionally, efficient optimizers are devised by machine learning experts based on theoretical insights [4, 15, 20, 11]. Adam [14] and its variant AdamW [20] remain the most widely employed methods in deep learning. Recently, however, a new optimization named Lion (Evolved Sign Momentum) [7] was discovered by an evolutionary search algorithm [32] applied to a symbolically represented program space [3]. Lion has been shown to achieve at least comparable performance to AdamW on a wide range of tasks while reducing memory cost and training time [7].

However, as the outcome of a stochastic search algorithm, Lion does not have an *a priori theoretical guarantee by design*. It is still uncertain whether Lion can be regarded as a reliable and legitimate general-purpose optimization algorithm, despite the reported positive results on a large, yet finite, set of tasks [7]. The lack of theoretical understanding also significantly restricts the potential for improving and extending Lion to obtain better new optimizers.

In this work, we demonstrate that Lion, along with a broader family of Lion-$\mathcal{K}$ algorithms, can be established as a theoretically novel and intriguing approach for solving optimization problems with convex regularization or constraints. This is surprising because Lion was discovered in a search

---

$^*$Equal Contribution

space that includes arbitrary symbolic operations and was not designed with any theoretical guarantees. This discovery opens up promising opportunities for developing improved optimizers by leveraging the existing success of Lion.

**Lion: Evolved Sign Momentum**   The update rule of Lion for minimizing a loss $f(x)$ on $\mathbb{R}^d$ is

$$\text{Lion:} \quad \begin{aligned} m_{t+1} &= \beta_2 m_t - (1 - \beta_2)\nabla f(x_t), \\ x_{t+1} &= x_t + \epsilon(\text{sign}(\beta_1 m_t - (1 - \beta_1)\nabla f(x_t)) - \lambda x_t), \end{aligned} \tag{1}$$

where $m_t \in \mathbb{R}^d$ is the momentum, $\epsilon > 0$ is the learning rate, $\beta_1, \beta_2 \in [0, 1]$ are two momentum related coefficients, and $\lambda \geq 0$ is a weight decay coefficient. A default value of $\beta_1 = 0.9$ and $\beta_2 = 0.99$ was suggested in Chen et al. [7], with which the Lion update rule can be written directly as

$$x_{t+1} \leftarrow (1 - \epsilon\lambda)x_t - \epsilon \, \text{sign}\left((10 + 1)g_t + 0.99g_{t-1} + 0.99^2 g_{t-2} + \cdots 0.99^k g_{t-k} + \cdots\right),$$

where $g_t = \nabla f(x_t)$. Here the update of $x_t$ combines a weight decay term with coefficient $(1 - \epsilon\lambda)$, and the sign of a weighted average of the trajectory gradients. Notably, the weight of the current gradient $g_t$ is increased by $(\beta_2 - \beta_1)/((1 - \beta_2)\beta_1) \approx 10$ times compared with typical exponential moving average of gradients as used in the classical Polyak momentum [30].

One can think of Lion as made by "splicing" the elements of many existing algorithms in Lion, which is exactly what an efficient search program can do when given a proper search space [29, 7, 3]. The update of the momentum $m_t$ is common to the Polyak momentum-based algorithms and yields the exponential moving average part of the update. What sets it apart is the unique update of $x_t$, which uses the combination of three key elements:

i) **[Sign Reshaper]** The use of the $\text{sign}(\cdot)$ function for update, similar to signed gradient descent and signed momentum [5, 8], can be viewed as an extreme way of normalizing the magnitude of the coordinate-wise updates. It is closed related to normalized gradient [19, 25] and adaptive gradient methods such as Adam [14] and RMSprop [36]. Note that Adam can be viewed as signed momentum with an adaptive variance based step size [2], which might be the key factor explaining the gap between Adam and SGD [18].

ii) **[Gradient Enhancement]** When using $\beta_2 > \beta_1$, the importance of the current gradient $g_t$ is increased compared to the exponential moving average in standard Polyak momentum update. It can be shown that Polyak momentum with this gradient enhancement results in Nesterov momentum, and leads to the well-known acceleration phenomenon [e.g., 35].

iii) **[Decoupled Weight Decay]** The weight decay term $\lambda x_t$ outside of the gradient and $\text{sign}(\cdot)$. Such idea of the *decoupled* weight decay is what make AdamW [21] significantly outperform the vanilla Adam in training large AI models.

As demonstrated by the empirical findings of Chen et al. [7] and subsequent research, the combination of these elements has been shown to make Lion perform well on a wide range of problems, including image classification, language models, and diffusion models [7].

However, it remains unclear whether the combination of these elements yield a theoretically valid and convergent general-purpose optimizer. Furthermore, the use of decoupled weight decay adds to the uncertainty regarding what optimization problem Lion aims to solve: due to its interaction with other parts of the algorithm, decoupled weight decay is always not equivalent to simply introducing $\ell_2$ regularization [20].

**"Lion King Meets Mr. Lyapunov"**   We propose and analyze a general family of Lion-$\mathcal{K}$ algorithms, in which we replace the $\text{sign}(\cdot)$ function in Lion with a subgradient $\nabla\mathcal{K}$ of a general convex function $\mathcal{K}: \mathbb{R}^d \to \mathbb{R}$:

$$\text{Lion-}\mathcal{K}: \quad \begin{aligned} m_{t+1} &= \beta_2 m_t - (1 - \beta_2)\nabla f(x_t), \\ x_{t+1} &= x_t + \epsilon(\nabla\mathcal{K}(\beta_1 m_t - (1 - \beta_1)\nabla f(x_t)) - \lambda x_t). \end{aligned} \tag{2}$$

Lion is recovered when $\mathcal{K}(x) = \|x\|_1$ and $\nabla\mathcal{K}(x) = \text{sign}(x)$. Taking the continuous time limit of (2), we obtain the following ordinary differential equation (ODE):

$$\text{Lion-}\mathcal{K} \text{ (ODE):} \quad \begin{aligned} \dot{m}_t &= -\alpha\nabla f(x_t) - \gamma m_t \\ \dot{x}_t &= \nabla\mathcal{K}(m_t - \varepsilon(\alpha\nabla f(x_t) + \gamma m_t)) - \lambda x_t, \end{aligned} \tag{3}$$

| | |
|---|---|
| Polyak Momentum [30] | $\mathcal{K}(x) = \|x\|_2^2/2, \gamma\lambda = 0, \varepsilon = 0$ |
| Nesterov Momentum [27] | $\mathcal{K}(x) = \|x\|_2^2/2, \gamma\lambda = 0$ |
| Signed Momentum [5] | $\mathcal{K}(x) = \|x\|_1^2, \varepsilon = 0, \lambda = 0$ |
| Hamiltonian Descent [22] | $\varepsilon = 0, \lambda = 0$ |
| Hamiltonian Descent for Composite Objectives [22] | $\varepsilon = 0, \lambda > 0$ |
| Dual Space Preconditioning [23], Mirror Descent [26] | $\varepsilon\gamma = 1, \lambda = 0$ |
| Signed Gradient Descent [5] | $\mathcal{K}(x) = \|x\|_1, \varepsilon\gamma = 1, \lambda = 0$ |
| Accelerated Mirror Descent [16] | $\gamma = 0, \varepsilon = 0, \lambda > 0$ |
| Frank–Wolfe [10] | $\varepsilon\gamma = 1, \lambda > 0$ |

Table 1: Lion-$\mathcal{K}$ includes a large family algorithms as special cases. See Section 3.1

Eq. (2) is the Euler discretization of Eq. (3) with step size $\epsilon$ in the case of $\alpha = \gamma$, with $\beta_1 = 1 - \varepsilon\gamma$, and $\beta_2 = 1 - \epsilon\gamma$. Lion-$\mathcal{K}$ includes a broad set of algorithms as special cases, as shown in Table 1.

To avoid the complexities associated with regularity conditions, we can assume that $\mathcal{K}$ is continuously differentiable when discussing the ODE. But parallel results hold for the time discrete algorithm (2) for general non-differentiable convex functions $\mathcal{K}$.

The crest of this work is to show that, when $\varepsilon\gamma \le 1$, Lion-$\mathcal{K}$ ODE solves the following optimization:

$$\min_{x\in\mathbb{R}^d} F(x) := \alpha f(x) + \frac{\gamma}{\lambda}\mathcal{K}^*(\lambda x), \tag{4}$$

where $\mathcal{K}^*(x) := \sup_z(x^\top z - \mathcal{K}(z))$ is the conjugate function of $\mathcal{K}$. Because we may have $\mathcal{K}^*(x) = +\infty$ for some $x$, solving (4) requires to enforce a constraint of $\lambda x \in \text{dom}\mathcal{K}^*$, where $\text{dom}\mathcal{K}^* := \{x: \mathcal{K}^*(x) < +\infty\}$ is the effective domain of $\mathcal{K}^*$. In the case of Lion, we have $\mathcal{K}(x) = \|x\|_1$ and hence $\mathcal{K}^*(x) = \delta(\|x\|_\infty \le 1)$, where $\delta$ the $\infty$-indicator function with $\delta(\texttt{True}) = 0, \delta(\texttt{False}) = +\infty$. Hence, Lion solves the following bound-constrained optimization problem:

$$\min_{x\in\mathbb{R}^d} f(x) \quad s.t. \quad \|x\|_\infty \le 1/\lambda, \tag{5}$$

where the bound $1/\lambda$ is solely decided by the weight decay coefficient $\lambda$.

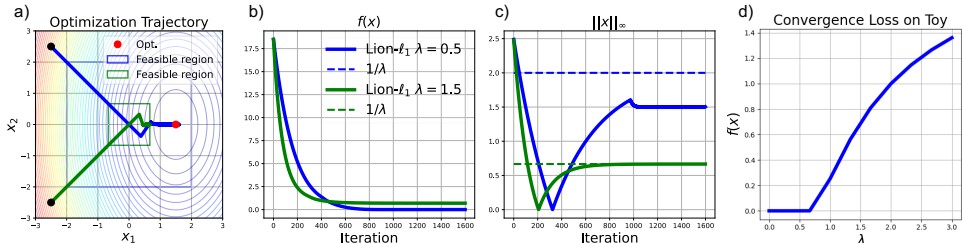

Figure 1: (a)-(c) Trajectories of Lion on 2D function $f(x) = (x_1 - 1.5)^2 + x_2^2$, with $\lambda = 1.5$ and $\lambda = 0.5$ ((a)-(c)). The boxes in a) represent the constraint set : blue box is for $\|x\|_\infty \le 1/\lambda$ with $\lambda = 0.5$, green box is for $\lambda = 1.5$. (d) $\lambda$ vs. the converged loss We can see that the converged loss starts to increase only when $\lambda$ excel a threshold ($\lambda \ge 0.6$) to excluded the unconstrained minimum from the constrained set.

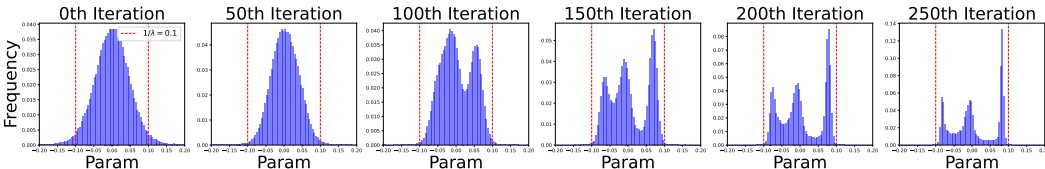

Figure 2: Histograms of the network parameters of ResNet-18 on CIFAR-10 trained by Lion with $\lambda = 10$. The constraint of $\|x\|_\infty \le 1/\lambda$ (indicated by the red vertical lines) is satisfied within only $\sim$200 steps.

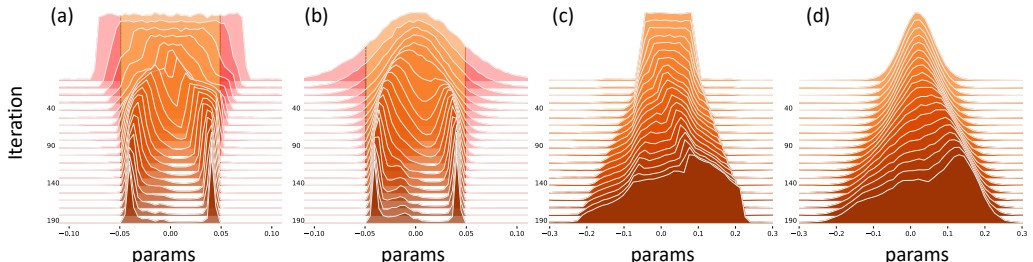

Figure 3: Evolution of histogram of parameter weights trained by Lion on ResNet-18 on CIFAR-10 [13, 17], with different $\lambda$ and initialization methods. Frequency of network parameters in ResNet on the CIFAR-10 dataset across iterations. (a): Kaiming uniform initialization [12] and $\lambda = 20$. (b): Kaiming normal initialization [12] and $\lambda = 20$. (c): Kaiming uniform initialization [12] and $\lambda = 0$. (d): Kaiming normal initialization [12] and $\lambda = 0$. The weights are quickly confined into the bound $[-0.05, 0.05]$ with $\lambda = 20$, while keep growing with zero weight decay ($\lambda = 0$).

Our proof shows that the Lion-$\mathcal{K}$ dynamics consists of two phases:

1) **[Phase 1]** When $\lambda x \notin \text{dom}\mathcal{K}^*$, it exponentially decays the distance from $\lambda x_t$ to the set $\text{dom}\mathcal{K}^*$:

$$\text{dist}(\lambda x_t, \text{dom}\mathcal{K}^*) \leq \exp(-\lambda(t - s)) \, \text{dist}(\lambda x_s, \text{dom}\mathcal{K}^*), \quad \forall s \leq t.$$

Hence, $\lambda x_t$ converges to $\text{dom}\mathcal{K}^*$ rapidly and stays within $\text{dom}\mathcal{K}^*$ once it arrived.

2) **[Phase 2]** After $\lambda x_t$ enters $\text{dom}\mathcal{K}^*$, the dynamics minimizes the finite valued objective $F(x)$. This is proved by showing that the Lion-$\mathcal{K}$ dynamics minimizes the following Lyapunov function:

$$H(x, m) = \alpha f(x) + \frac{\gamma}{\lambda}\mathcal{K}^*(\lambda x) + \frac{1 - \varepsilon\gamma}{1 + \varepsilon\lambda}(\mathcal{K}^*(\lambda x) + \mathcal{K}(m) - \lambda m^\top x). \tag{6}$$

We show that, whenever $H(x_t, m_t)$ is finite, it is decreased monotonically (i.e., $\frac{\mathrm{d}}{\mathrm{d}t}H(x_t, m_t) \leq 0$) along trajectories of (3) until a local minimum of point of $H(x, m)$ is reached.

Furthermore, we have $F(x) = \min_m H(x, m)$, and hence minimizing $H(x, m)$ is equivalent to minimizing $F(x)$; this is because the minimum of the last term in (6) equals zero, $\min_m \mathcal{K}^*(\lambda x) + \mathcal{K}(m) - \lambda m^\top x = 0$, for any fixed $x$, by Fenchel-Young inequality.

The discovery of this Lyapunov function is a new and non-trivial mathematical result. But intuitively, one can see easily the connection of (3) and (4) by comparing their fixed points. Assume $\mathcal{K}$ and $\mathcal{K}^*$ are differentiable, then a fix point of (3) must implies a stationary point of (4):

$$\underbrace{\alpha\nabla f(x_t) + \gamma m_t = 0, \quad \nabla\mathcal{K}(m_t) = \lambda x_t}_{\text{fixed point of (3)}} \quad \Longrightarrow \quad \underbrace{\alpha\nabla f(x_t) + \gamma\nabla\mathcal{K}^*(\lambda x_t) = 0,}_{\text{stationary point of (4)}}$$

where we used $\nabla\mathcal{K}(\nabla\mathcal{K}^*(x)) = x$, and $\nabla_x \left(\frac{1}{\lambda}\mathcal{K}^*(\lambda x)\right) = \nabla\mathcal{K}^*(\lambda x)$.

**Why Should Lion Decay Weight?** From the analysis above, the role of weight decay $\lambda$ in Lion is two-fold:

1) It alternates the solution if $\lambda$ is large and the constraint $\|x\|_\infty \leq 1/\lambda$ is strong enough to exclude the unconstrained minimum $x_{\text{unc}}^*$ of $f(x)$. This may improve the generalization and stability of the solution while sacrificing the training loss.

2) If $\lambda$ is sufficiently small to include the unconstrained minimum $x_{\text{unc}}^*$ in the constrained set, it does not alter the final solution. In this case, the main role of weight decay is to speed up the convergence because Phase 1 brings the solution into the constrained set with a linear rate. Hence, the ideal choice of $\lambda$ is $\lambda = 1/\|x_{\text{unc}}^*\|_\infty$.

In Figure 4 we plot Lion's performance with different $\lambda$. The right plot confirms that larger $\lambda$ results in faster convergence but might sacrifice the performance. The left plot shows that there exists an optimal $\lambda$ (=0.56), beyond which the training loss starts to increase.

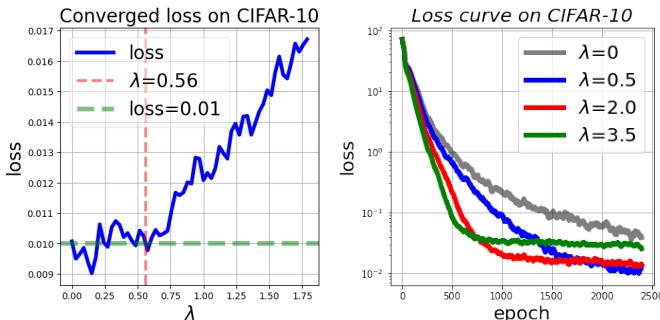

Figure 4: Analysis of weight decay on CIFAR-10 using Lion. a) The converged Loss vs. weight decay in Lion. We can see that the loss starts to increase only when $\lambda$ excel a threshold, which is expected from the constrained optimization view. b) The loss curves vs. epochs with different weight decays. Larger weight decay $\lambda$ yields faster convergence (due to stronger Phase 1), but may yield larger final loss when it is too large.

| Line ID | $\mathcal{K}(x)$ | $\nabla\mathcal{K}(x)$ | $\min_x f(x) + \mathcal{K}^*(x)$ |
|---|---|---|---|
| ① | $\|x\|_1$ | $\mathrm{sign}(x)$ | $\min f(x)$ $s.t.$ $\|x\|_\infty \leq 1$ |
| ② | $\|x\|_p$ | $\frac{\mathrm{sign}(x)|x|^{p-1}}{\|x\|_p^{p-1}}$ | $\min f(x)$ $s.t.$ $\|x\|_q \leq 1$ |
| ③ | $\sum_i \max(|x_i| - e, 0)$ | $\mathrm{sign}(x)\mathbb{I}(|x| > e)$ | $\min f(x) + e\|x\|_1$ $s.t.$ $\|x\|_\infty \leq 1$ |
| ④ | $\sum_{i \leq i^{cut}} |x_{(i)}|$ | $\mathrm{sign}(x)\mathbb{I}(|x| > |x_{(i^{cut})}|)$ | $\min f(x)$ $s.t.$ $\|x\|_1 \leq i^{cut}$, $\|x\|_\infty \leq 1$ |
| ⑤ | $\sum_i \mathrm{huber}_e(x_i)$ | $\mathrm{clip}(x, -e, e)/e$ | $\min f(x) + \frac{e}{2}\|x\|_2^2$ $s.t.$ $\|x\|_\infty < 1$ |

Table 2: Examples of $\mathcal{K}$ and $\nabla\mathcal{K}$, and the optimization problems they solved (we set $\gamma = \lambda = 1$ for simplicity). We assume $x = [x_1, \ldots, x_d] \in \mathbb{R}^d$ and $|x_{(1)}| \geq |x_{(2)}| \geq \cdots$ is a monotonic sorting of the elements of $x$, and $i^{cut}$ is an integer in $\{1, \ldots, d\}$. The Huber loss is $\mathrm{huber}_e(x_i) = \mathbb{I}(|x_i| \geq e)(|x_i| - \frac{e}{2}) + \mathbb{I}(|x_i| < e)\frac{1}{2e}x_i^2$, $e > 0$. See Appendix A for more examples.

**Going Beyond Lion**  Different $\mathcal{K}$ yield optimization with different convex constraints and/or regularizations. For example, using the $\ell_p$ norm $\mathcal{K}(x) = \|x\|_p$ yields a constraint on the dual norm $\|x\|_q \leq 1/\lambda$ where $1/p + 1/q = 1$ (Table 2, Line ②); zeroing out the coordinates with small magnitude corresponds to introducing an $\ell_1$ regularization (Line ③) or $\ell_1$ constraint (④), which is useful for sparse learning; replacing $\nabla\mathcal{K}(x) = \mathrm{sign}(x)$ with a continuous function would introduce an extra regularization term on the loss (e.g., ⑤). This work will focus on building the basic theoretical framework, and leave the vast opportunities of practical applications as future directions.

**Outline**  The rest of the paper is organized as follows. Section 2 introduces preliminaries on convex functions. Section 3 analyzes the continuous-time Lion-$\mathcal{K}$ dynamics and discusses connections with existing algorithms. Section 4 presents the discrete-time analysis. Section 5 presents experiments that study and verify the behavior of using different $\mathcal{K}$s.

## 2  PRELIMINARIES ON CONVEX FUNCTIONS

Assume $\mathcal{K}: \mathbb{R}^d \to \mathbb{R}$ is convex. A vector $u \in \mathbb{R}^d$ is said to be a subgradient of $\mathcal{K}$ at $x$, denoted as $u \in \partial\mathcal{K}(x)$, if

$$\mathcal{K}(y) - \mathcal{K}(x) \geq u^\top(y - x), \quad \forall y \in \mathbb{R}^d.$$

With an abuse of notation, we use $\nabla\mathcal{K}(x)$ to denote a subgradients of $\mathcal{K}$, that is, $\nabla\mathcal{K}(x) \in \partial\mathcal{K}(x)$. When $\mathcal{K}$ is differentiable at $x$, there is an unique subgradient $\nabla\mathcal{K}(x)$ which coincides with the regular derivative.

The conjugate function $\mathcal{K}^*$ of $\mathcal{K}$ is defined as

$$\mathcal{K}^*(x) = \sup_{z \in \mathbb{R}^d} (x^\top z - \mathcal{K}(z)).$$

Hence, by definition, we have the following Fenchel-Young inequality:

$$\mathcal{K}(x) + \mathcal{K}^*(y) \geq x^\top y, \quad \forall x, y. \tag{7}$$

The conjugate function $\mathcal{K}^*$ can take values in the extended real set $\overline{\mathbb{R}} = \mathbb{R} \cup \{\pm\infty\}$, and $\mathcal{K}^*$ is always closed and convex, even when $\mathcal{K}$ is not. Recall that a function $f$ is said to be closed if for each $b \in \mathbb{R}$, its sublevel sets $\{x \colon f(x) \leq b\}$ is a closed set.

If $\mathcal{K}$ is closed and convex, we have $\mathcal{K}^{**} = \mathcal{K}$, and

$$y \in \partial\mathcal{K}(x) \qquad \Longleftrightarrow \qquad x \in \partial\mathcal{K}^*(y) \qquad \Longleftrightarrow \qquad \mathcal{K}(x) + \mathcal{K}^*(y) = x^\top y. \tag{8}$$

When $\mathcal{K}$ and $\mathcal{K}^*$ are differentiable, (8) suggests that $\nabla\mathcal{K}$ and $\nabla\mathcal{K}^*$ is a pair of inverse maps: $\nabla\mathcal{K}(\nabla\mathcal{K}^*(x)) = x$. Combining (7) and (8), we get $\min_m \mathcal{K}(m) + \mathcal{K}^*(x) - x^\top m = 0$, which yields $F(x) = \min_m H(x, m)$. We refer to Rockafellar [33] for a systematic introduction to convex functions.

A key property of any subgradient $\nabla\mathcal{K}$ and $\nabla\mathcal{K}^*$ is that they are monotonic maps, which plays a crucial rule in our results.

**Lemma 2.1.** *Assume $\mathcal{K}, \mathcal{K}^*$ is a closed convex conjugate pair and $\nabla\mathcal{K}, \nabla\mathcal{K}^*$ are their subgradients, we have*

$$(\nabla\mathcal{K}(x) - \nabla\mathcal{K}(y))^\top(x - y) \geq 0, \qquad (\nabla\mathcal{K}(x) - y)^\top(x - \nabla\mathcal{K}^*(y)) \geq 0. \tag{9}$$

See Appendix B.1 for the proof. These two inequalities are crucial because they allow us to identify vectors that have a non-negative inner product with a given direction to achieve monotonic descent in optimization.

**Example 2.2.** *In the case of Lion, we take $\mathcal{K}(x) = \|x\|_1$ with $\nabla\mathcal{K}(x) = \mathrm{sign}(x)$, and*

$$\mathcal{K}^*(y) = \begin{cases} 0 & \text{if } \|y\|_\infty \leq 1 \\ +\infty & \text{if } \|y\|_\infty > 1 \end{cases}, \qquad [\nabla\mathcal{K}^*(y)]_i = \begin{cases} 0 & \text{if } |y_i| \leq 1 \\ +\infty & y_i > 1 \\ -\infty & y_i < -1. \end{cases}$$

*One can verify that the inequalities in (9) hold (even though the values on the left side can be $+\infty$). The Lyapunov function in (6) becomes*

$$H(x, m) = \begin{cases} f(x) + \frac{1-\varepsilon\gamma}{1+\varepsilon\lambda}(\|m\|_1 - \lambda x^\top m) & \text{if } \|x\|_\infty \leq 1 \\ +\infty & \text{if } \|x\|_\infty > 1. \end{cases}$$

## 3   MAIN RESULT: CONTINUOUS-TIME

We study the continuous-time Lion-$\mathcal{K}$ dynamics (3), and discuss its connection to existing algorithms listed in Table 1. We defer the detailed proofs to Appendix B.7, but outline a novel *implicit Hamiltonian + descent decomposition* that underpins the construction of the Lyapunov function $H(x, m)$.

**Theorem 3.1.** *Let $(x_t, m_t)$ be a continuously differentiable trajectory of the Lion-$\mathcal{K}$ ODE (3), where $\mathcal{K}$ is differentiable convex with conjugate $\mathcal{K}^*$. Assume $\alpha, \gamma, \lambda, \varepsilon > 0$ and $\epsilon\gamma \leq 1$.*

*1) [Phase 1] Define $\mathrm{dist}(\lambda x_t, \mathrm{dom}\mathcal{K}^*) = \inf_{z \in \mathrm{dom}\mathcal{K}^*} \|z - \lambda x_t\|$ w.r.t. any norm $\|\cdot\|$. We have*

$$\mathrm{dist}(\lambda x_t, \mathrm{dom}\mathcal{K}^*) \leq \exp(\lambda(s-t)) \, \mathrm{dist}(\lambda x_s, \mathrm{dom}\mathcal{K}^*), \quad \forall 0 \leq s \leq t.$$

*Hence, $\lambda x_t$ converges linearly to set $\mathrm{dom}\mathcal{K}^*$ and stays within $\mathrm{dom}\mathcal{K}^*$ once it enters it.*

*2) [Phase 2] When $H(x, m)$ in (6) is finite and continuously differentiable, it is decreased monotonically along the trajectory:*

$$-\frac{\mathrm{d}}{\mathrm{d}t}H(x_t, m_t) = \Delta(x_t, m_t) := \frac{\lambda + \gamma}{1 + \varepsilon\lambda}\Delta_1(x_t, \tilde{m}_t) + \frac{1 - \varepsilon\gamma}{1 + \varepsilon\lambda}\Delta_2(m_t, \tilde{m}_t) \geq 0,$$

*where we define $\tilde{m}_t = m_t - \varepsilon(\alpha\nabla f(x_t) + \gamma m_t)$, and*

$$\begin{aligned}
\Delta_1(x_t, \tilde{m}_t) &= (\tilde{m}_t - \nabla\mathcal{K}^*(\lambda x_t))^\top(\nabla\mathcal{K}(\tilde{m}_t) - \lambda x_t) \geq 0, \\
\Delta_2(m_t, \tilde{m}_t) &= \frac{1}{\varepsilon}(\tilde{m}_t - m_t)^\top(\nabla\mathcal{K}(\tilde{m}_t) - \nabla\mathcal{K}(m_t)) \geq 0.
\end{aligned} \tag{10}$$

*3) [Stationarity] Assume $\nabla \mathcal{K}^*$ is strictly monotonic. All the accumulation points of $(x_t, m_t)$ as $t \to +\infty$ are stationary points of the objective function $F(x) = \alpha f(x) + \frac{\gamma}{\lambda} \mathcal{K}^*(\lambda x)$, and satisfy $\lambda x \in \text{dom} \mathcal{K}^*$.*

$\Delta(x_t, m_t)$ can be viewed as an indication of the stationarity of the system. If $H(x_0, m_0)$ is finite and $H_b := \inf_{x,m} H(x, m) > -\infty$, we have $\frac{1}{T} \int_0^T \Delta(x_t, m_t) \mathrm{d}t \leq \frac{H(x_0, m_0) - H_b}{T} \to 0$ when $T \to +\infty$.

*Proof Sketch.* See Appendix B.7 for the full proof. The original discovery of the Lyapunov function was made possible by starting from the inequalities in (10) as guaranteed by Lemma 2.1, and working backwards with some guesswork. The following is a simplified proof that highlights the essential mathematical structure that makes $H(x, m)$ Lyapunov. Define

$$\dot{x} = V_x(x, m) := \nabla \mathcal{K}(\tilde{m}) - \lambda x, \qquad \dot{m} = V_m(x, m) := -\alpha \nabla f(x) - \gamma m = \frac{\tilde{m} - m}{\varepsilon}$$

and related

$$\hat{V}_x(x, m) = \tilde{m} - \nabla \mathcal{K}^*(\lambda x), \qquad \hat{V}_m(x, m) = \nabla \mathcal{K}(\tilde{m}) - \nabla \mathcal{K}(m).$$

The $\hat{V}_x$ and $\hat{V}_m$ have two critical properties:

1) By Lemma 2.1, $\hat{V}_x$ and $\hat{V}_m$ have non-negative inner products with $V_x, V_m$, respectively:

$$\hat{V}_x(x, m)^\top V_x(x, m) \geq 0, \qquad \hat{V}_m(x, m)^\top V_m(x, m) \geq 0, \qquad \forall x, m.$$

2) By Lemma B.5 in Appendix B.7, the gradients of $H$ can be decomposed as follows:

$$\begin{aligned} \nabla_x H(x, m) &= -\eta' \hat{V}_x - \eta V_m \\ \nabla_m H(x, m) &= -\eta \hat{V}_m + \eta V_x, \end{aligned} \qquad \textbf{(Implicit Hamiltonian + Descent)} \qquad (11)$$

where $\eta = \frac{1 - \varepsilon\gamma}{1 + \varepsilon\lambda}$ and $\eta' = \frac{\gamma + \lambda}{1 + \varepsilon\lambda}$. We call (11) an *"implicit" Hamiltonian + descent* decomposition, in connection with the Hamiltonian + descent decomposition we introduce in sequel.

Then we have,

$$\frac{\mathrm{d}}{\mathrm{d}t} H(x_t, m_t) = \nabla_x H^\top V_x + \nabla_m H^\top V_m = (-\eta' \hat{V}_x - \eta V_m)^\top V_x + (-\eta \hat{V}_m + \eta V_x)^\top V_m$$

$$= -(\eta' \hat{V}_x^\top V_x + \eta \hat{V}_m^\top V_m) \leq 0.$$

The key here is that the cross term $\eta V_x^\top V_m$ is canceled, leaving only the negative terms. The convergence property uses Lasselle's invariance principle; see Appendix B.7 for details. $\qquad \square$

**Hamiltonian + Descent Decomposition** The decomposition structure (11) is a key characterization of Lion-$\mathcal{K}$ ODE. An interesting remark is that $H(x, m)$ is also Lyapunov if we have the following *Hamiltonian + descent* structure [22, 28] in which the roles of $[\nabla_x H, \nabla_m H]$ and $[V_x, V_m]$ in (11) are switched:

$$\begin{aligned} V_x &= -\hat{H}_x - \eta \nabla_m H \\ V_m &= -\hat{H}_m + \eta \nabla_x H, \end{aligned} \qquad \textbf{(Hamiltonian + Descent)} \qquad (12)$$

where $\hat{H}_x, \hat{H}_m$ are two vector fields satisfying $\hat{H}_x^\top (\nabla_x H) \geq 0$ and $\hat{H}_m^\top (\nabla_m H) \geq 0$, then

$$\frac{\mathrm{d}}{\mathrm{d}t} H(x_t, m_t) = \nabla_x H^\top V_x + \nabla_m H^\top V_m = \nabla_x H^\top (-\hat{H}_x - \eta \nabla_m H) + \nabla_m H^\top (-\hat{H}_m + \eta H_x)$$

$$= -(\hat{H}_x^\top (\nabla_x H) + \hat{H}_m^\top (\nabla_m H)) \leq 0.$$

The structure in (12) can be intuitively viewed as a generalized damped Hamiltonian system with $H(x, m)$ as the total energy, where $[-\hat{H}_x, -\hat{H}_m]$ serves a damping force that monotonically decreases the total energy, and $[-\nabla_m H, \nabla_x H]$ is the Hamiltonian vector field which preserves the energy but introduces an inertia-like effect into system. One can easily verify (12) on the classical Polayk's momentum. The more general idea is explored in the Hamiltonian descent method of [22, 28], which considers systems of structure (12) for the separatiable Hamiltonian of form $H(x, m) = f(x) + \mathcal{K}(m)$ with $\hat{H}_x = 0$. In contrast, (11) do not seem to have a clear physical interpretation, yet provides a handy tool for understanding the general Lion-$\mathcal{K}$ dynamics. Some special cases of Lion-$\mathcal{K}$, such as when $\lambda = 0$ or $\varepsilon = 0$, can also be alternatively viewed from the Hamiltonian + descent structure as shown in Section 3.1.

## 3.1 Connection with Existing Algorithms

What makes Lion-$\mathcal{K}$ unique is the combination of the gradient enhancement ($\varepsilon > 0$), the decoupled weight decay ($\lambda > 0$), and the momentum damping ($\gamma > 0$), the use of reshaper function $\nabla\mathcal{K}(\cdot)$. We discuss the effects of these elements in connection to existing algorithms as shown in Table 1.

**Lion-$\mathcal{K}$ Without Weight Decay**  When $\lambda = 0$ and $\nabla\mathcal{K}^*(0) = 0$, we have $\lim_{\lambda\to 0}\frac{1}{\lambda}\mathcal{K}^*(\lambda x) = \nabla\mathcal{K}(0)^\top x = 0$, and the Lyapunov function can be defined as

$$H(x, m) = \alpha f(x) + (1 - \varepsilon\gamma)\mathcal{K}(m),$$

for which we have

$$-\frac{\mathrm{d}}{\mathrm{d}t}H(x_t, m_t) = \gamma\nabla\mathcal{K}(\tilde{m}_t)\tilde{m}_t + \frac{(1 - \varepsilon\gamma)}{\varepsilon}(\tilde{m}_t - m_t)^\top(\nabla\mathcal{K}(\tilde{m}_t) - \nabla\mathcal{K}(m_t)) \geq 0.$$

In this case, the algorithm solves $\min_x f(x)$, without the regularization term $\mathcal{K}^*(\lambda x)$.

Interestingly, in this case ($\lambda = 0$) and $1 - \varepsilon\gamma > 0$, there exists a second Lyapunov function:

$$\tilde{H}(x, m) = \alpha f(x) + \frac{1}{1 - \varepsilon\gamma}\mathcal{K}((1 - \varepsilon\gamma)m), \tag{13}$$

with which the Lion-$\mathcal{K}$ ODE ($\lambda = 0$) can be decomposed in the form of (12), as a sum of a Hamiltonian vector field and a descent direction:

$$\begin{bmatrix} \dot{x}_t \\ \dot{m}_t \end{bmatrix} = \underbrace{\begin{bmatrix} +\nabla_m\tilde{H}(x_t, m_t) \\ -\nabla_x\tilde{H}(x_t, m_t) \end{bmatrix}}_{\text{Hamiltonian}} - \underbrace{\begin{bmatrix} \nabla\mathcal{K}(\tilde{m}_t^0) - \nabla\mathcal{K}(\tilde{m}_t) \\ \gamma m_t \end{bmatrix}}_{\text{Descent}},$$

where $\tilde{m}_t^0 = (1 - \varepsilon\gamma)m_t$ and hence $\tilde{m}_t^0 - \tilde{m}_t = \varepsilon\alpha\nabla f(x_t)$. If $m = 0$ is a minimum of $\mathcal{K}(m)$, one can show that the second component above is a descent direction of $\tilde{H}(x, m)$ in (13), with

$$-\frac{\mathrm{d}}{\mathrm{d}t}\tilde{H}(x_t, m_t) = \gamma\nabla\mathcal{K}(\tilde{m}_t^0)^\top m_t + \frac{1}{\varepsilon}(\tilde{m}_t^0 - \tilde{m}_t)^\top(\nabla\mathcal{K}(\tilde{m}_t^0) - \nabla\mathcal{K}(\tilde{m}_t)) \geq 0,$$

See Appendix B.6 for details.

**Lion-$\mathcal{K}$ Without Momentum Damping**  When $\gamma = 0$, we have

$$H(x, m) = \alpha f(x) + \frac{1}{1 + \varepsilon\lambda}(\mathcal{K}^*(x) + \mathcal{K}(m) - \lambda x^\top m),$$

Because $\min_m(\mathcal{K}^*(x) + \mathcal{K}(m) - \lambda x^\top m) = 0$, the algorithm also corresponds to solving $\min_x f(x)$ without regularization $\mathcal{K}^*(\lambda x)$.

It is interesting to see that the weight decay and momentum damping play a somewhat symmetric role, because turning off either one of it turns off the regularization term $\mathcal{K}^*(\lambda x)$. In particular, if $\mathcal{K}(x) = \|x\|_2^2/2$, the Lion-$\mathcal{K}$ ODE can be rewritten into a second-order ODE:

$$\ddot{x}_t + (\lambda + \gamma)\dot{x}_t + \varepsilon\alpha\nabla^2 f(x_t)\dot{x}_t + \gamma\lambda x_t + \alpha\nabla f(x_t) = 0, \tag{14}$$

in which the role of $\gamma, \lambda$ are symmetric. Equation (21) coincides the high-resolution ODE in [35] for minimizing $F(x) = \alpha f(x) + \gamma\lambda\|x\|_2^2/2$, which is a high resolution continuous time limit of Nesterov momentum. The hessian-based damping term $\nabla^2 f(x_t)\dot{x}_t$ plays a key role for acceleration phenomenon [see e.g., 35, 1]. When we turn off the gradient enhancement ($\varepsilon = 0$), then we get ODE for Ployak momentum.

Interestingly, if we set $\lambda = \gamma = 0$, but $\varepsilon > 0$, ODE (21) still serve to minimize $f(x)$, due to the Hessian damping term.

**Lion-$\mathcal{K}$ without Gradient Enhancement**  When $\varepsilon = 0$, we have

$$H(x, m) = \alpha f(x) + \frac{\gamma}{\lambda}\mathcal{K}^*(\lambda x) + (\mathcal{K}^*(\lambda x) + \mathcal{K}(m) - \lambda m^\top x),$$

and $\Delta_2(m, \tilde{m}) = 0$,

$$\Delta(x, m) = (\lambda + \gamma)\Delta_1(x, m) = (\lambda + \gamma)(m - \nabla\mathcal{K}^*(\lambda x))^\top(\nabla\mathcal{K}(m) - \lambda x).$$

In this case, minimizing $H(x, m)$ still yields the minimization of $F(x)$. Hence, the choice of $\varepsilon$ does not alter the objective function.

Moreover, with $\varepsilon = 0$, one can conveniently decompose the velocity field in the form of (12), as a sum of a Hamiltonian vector field and mirror descent direction:

$$\begin{bmatrix} \dot{x}_t \\ \dot{m}_t \end{bmatrix} = \underbrace{\begin{bmatrix} +\nabla_m H(x_t, m_t) \\ -\nabla_x H(x_t, m_t) \end{bmatrix}}_{\text{Hamiltonian}} - \underbrace{\begin{bmatrix} 0 \\ (\gamma + \lambda)(m_t - \nabla\mathcal{K}^*(\lambda x_t)) \end{bmatrix}}_{\text{Descent}}.$$

This system can be shown to be equivalent to the Hamiltonian descent system for composite objects of [28]. Further, if $\lambda = 0$, it reduces to the conformal Hamiltonian system [e.g., 22, 24].

**Mirror Descent and Frank-Wolfe**  If $\varepsilon\gamma = 1$, Lion-$\mathcal{K}$ reduces to

$$\dot{x}_t = \nabla\mathcal{K}(-\varepsilon\alpha\nabla f(x_t)) - \lambda x_t,$$

which can be shown to be equivalent to the Frank-Wolfe algorithm for minimizing $F(x) = \alpha f(x) + \frac{\gamma}{\lambda}\mathcal{K}^*(\lambda x)$.

When $\varepsilon\gamma = 1$, and $\lambda = 0$ with $\nabla\mathcal{K}(x) = 0$ iff $x = 0$, Lion-$\mathcal{K}$ reduces to $\dot{x}_t = \nabla\mathcal{K}(-\varepsilon\alpha\nabla f(x_t))$, which is dual space conditioning [23], or a variant of mirror descent for $\min_x f(x)$. See Appendix B.4 for more discussion.

**Accelerated Mirror Descent**  The accelerated mirror descent of Krichene et al. [16] is

$$\dot{x}_t = \lambda_t(\nabla\mathcal{K}(m_t) - x_t), \qquad\qquad \dot{m}_t = -\alpha_t\nabla f(x_t),$$

which is shown to exhibit an acceleration behavior for minimizing a convex $f$ (without the $\mathcal{K}^*$ regularization) when $\alpha_t = t/r$ and $\lambda_t = r/t$ and $r \geq 2$. This can be viewed as Lion-$\mathcal{K}$ ODE with $\gamma = 0, \varepsilon = 0$ and but a special time-dependent coefficient.

## 4  DISCRETE TIME ANALYSIS

We now present a result on the discrete-time Lion-$\mathcal{K}$ parallel to the continous-time results in Theorem 3.1, but work for non-differentiable convex functions $\mathcal{K}$. We analyze a slight reform of (2):

$$\begin{aligned} m_{t+1} &= \beta_2 m_t - (1 - \beta_2)\nabla f(x_t) \\ \tilde{m}_{t+1} &= \beta_1 m_t - (1 - \beta_1)\nabla f(x_t) \\ x_{t+1} &= x_t + \epsilon(\nabla\mathcal{K}(\tilde{m}_{t+1}) - \lambda x_{t+1}), \end{aligned} \tag{15}$$

in which we use an implicit scheme for the $x_t$-update, replacing $\lambda x_t$ with $\lambda x_{t+1}$. It is equivalent to the explicit scheme in (2) with $\epsilon$ replaced by $\epsilon' = \frac{\epsilon}{1+\epsilon\lambda}$.

**Theorem 4.1.** *Assume $f\colon \mathbb{R}^d \to \mathbb{R}$ is L-smooth, and $\mathcal{K}\colon \mathbb{R}^d \to \mathbb{R}$ is closed and convex, and $\nabla\mathcal{K}$ is a subgradient of $\mathcal{K}$. Assume $\beta_1, \beta_2 \in (0, 1)$, and $\beta_2 > \beta_1$, and $\epsilon, \lambda > 0$.*

*1) For any two non-negative integers $s \leq t$, we have*

$$\mathrm{dist}(\lambda x_t, \mathrm{dom}\mathcal{K}^*) \leq \left(\frac{1}{1+\epsilon\lambda}\right)^{s-t} \mathrm{dist}(\lambda x_s, \mathrm{dom}\mathcal{K}^*), \quad \forall s \leq t.$$

*2) Define the following Lyapunov function:*

$$H(x, m) = f(x) + \frac{1}{\lambda}\mathcal{K}^*(\lambda x) + \frac{\beta_1}{\epsilon\lambda(1 - \beta_1) + (1 - \beta_2)}(\mathcal{K}^*(\lambda x) + \mathcal{K}(m) - \lambda x^\top m),$$

*and*

$$\Delta_t^1 = (\nabla \mathcal{K}(\tilde{m}_{t+1}) - \lambda x_{t+1})^\top (\tilde{m}_{t+1} - \nabla \mathcal{K}^*(\lambda x_{t+1})) \geq 0,$$

$$\Delta_t^2 = (\nabla \mathcal{K}(\tilde{m}_{t+1}) - \nabla \mathcal{K}(m_{t+1}))^\top (\tilde{m}_{t+1} - m_{t+1}) \geq 0,$$

*where $\nabla \mathcal{K}^*$ is a subgradient of $\mathcal{K}^*$. Then we have*

$$H(x_{t+1}, m_{t+1}) - H(x_t, m_t) \leq -\epsilon \Delta_t + \frac{L\epsilon^2}{2} \|\nabla \mathcal{K}(\tilde{m}_{t+1}) - \lambda x_{t+1}\|_2^2,$$

*where $\Delta_t = a\Delta_t^1 + b\Delta_t^2$, with*

$$a = \frac{\beta_1}{\epsilon\lambda(1-\beta_1) + (1-\beta_2)} + 1 \geq 0, \qquad b = \frac{\beta_1(1-\beta_2)}{\epsilon\lambda(\beta_2-\beta_1)(\epsilon\lambda(1-\beta_1) + (1-\beta_2))} \geq 0.$$

*Hence, a telescoping sum yields*

$$\frac{1}{T}\sum_{t=0}^{T-1} \Delta_t \leq \frac{H(x_0, m_0) - H(x_T, m_T)}{\epsilon T} + \frac{L\epsilon}{2} B_T,$$

*where $B_T = \frac{1}{T}\sum_{t=1}^{T} \|\nabla \mathcal{K}(\tilde{m}_{t+1}) - \lambda x_{t+1}\|_2^2$.*

The result above shows that $\frac{1}{T}\sum_{t=0}^{T-1} \Delta_t$ decays with an $O(\frac{1}{\epsilon T} + \epsilon)$ rate, if $B_T$ is a finite upper bound. This reduces to the continuous-time result of $\frac{1}{t}\int_0^t \Delta(x_s, m_s)\mathrm{d}s = O\left(\frac{1}{t}\right)$ when the step size $\epsilon$ converges to zero.

If $\mathcal{K}$ is smooth, it is possible to improve the discrete-time rate to $O\left(\frac{1}{\epsilon T}\right)$ with standard arguments based on the proof of Theorem 4.1. Hence, the impact of the non-differentiability of $\mathcal{K}$ contributes to the $O(\epsilon)$ term, which suggests that the algorithm converges upto an $\epsilon$ accuracy. This is an typical phenomenon in optimization with non-smooth objectives (like sub-gradient descent) or non-smooth update (like signed GD). Because in practice the step size is small or decaying, the $O(\epsilon)$ term may not have a substantial impact for practical performance.

## 5 EXPERIMENTS ON DIFFERENT $\mathcal{K}$

This section provides a preliminary investigation on the behaviors of Lion-$\mathcal{K}$ with different $\mathcal{K}$. We experiment with the $\mathcal{K}$s listed in Table 2 on the toy example shown in Figure 1 to confirm the behavior follows exactly as what the theory predicts. Then we focus on the Lion-$\ell_p$ optimizer with general $p \in [1, 2]$ since it is the most straightforward extension of the original Lion (with $p = 1$).

### 5.1 LION-$\mathcal{K}$S ON THE TOY EXAMPLE

In the following, we plot the behavior of different Lion-$\mathcal{K}$s on the toy example shown in Figure 1. For each $\mathcal{K}$, we draw the optimization trajectory using the corresponding optimizer, the loss $f(x)$, and the corresponding constraint (e.g., the norm of $x$) v.s. iteration. The results are shown in Figure 5.

**Observation** From Figure 5, one can observe that for $\mathcal{K}(x) = \|x\|_2$, the constraint is a circle. For $\mathcal{K}(x) = \sum_i \max(|x_i| - e, 0)$, an additional $\ell_1$ regularization is introduced in addition to the $\ell_\infty$ constraint, which encourages sparse solutions. When $\mathcal{K}(x) = \sum_{i \leq i^{cut}} |x_{(i)}|$, it enforces an $\ell_1$ constraint (rather than regularization) in addition to the $\ell_\infty$ constraint. The $\mathcal{K}(x) = \sum_i \mathrm{huber}_e(x_i)$ introduces an $\ell_2$ regularization effect in addition to $\ell_\infty$ constraint. All optimization trajectories closely match what the theory predicts.

### 5.2 LION-$\ell_p$ FOR IMAGENET AND LANGUAGE MODELING

Lion-$\ell_p$ corresponds to $\mathcal{K}(x) = \|x\|_p$, $p \geq 1$ and amounts to solving $\min_x f(x)$ $s.t.$ $\|x\|_q \leq 1/\lambda$ where $1/p + 1/q = 1$. In Figure 6, we plot how the parameter norms (e.g., $\|\cdot\|_\infty$ when $p = 1$ and $\|\cdot\|_2$ when $p = 2$) change over training iterations. In Figure 7, we compare the performance of using Lion-$\ell_p$ with different $p$, on ImageNet [34] and Language Modeling tasks, using ResNet-50, Vision Transformer (ViT) [9], and the GPT-2 model [31].

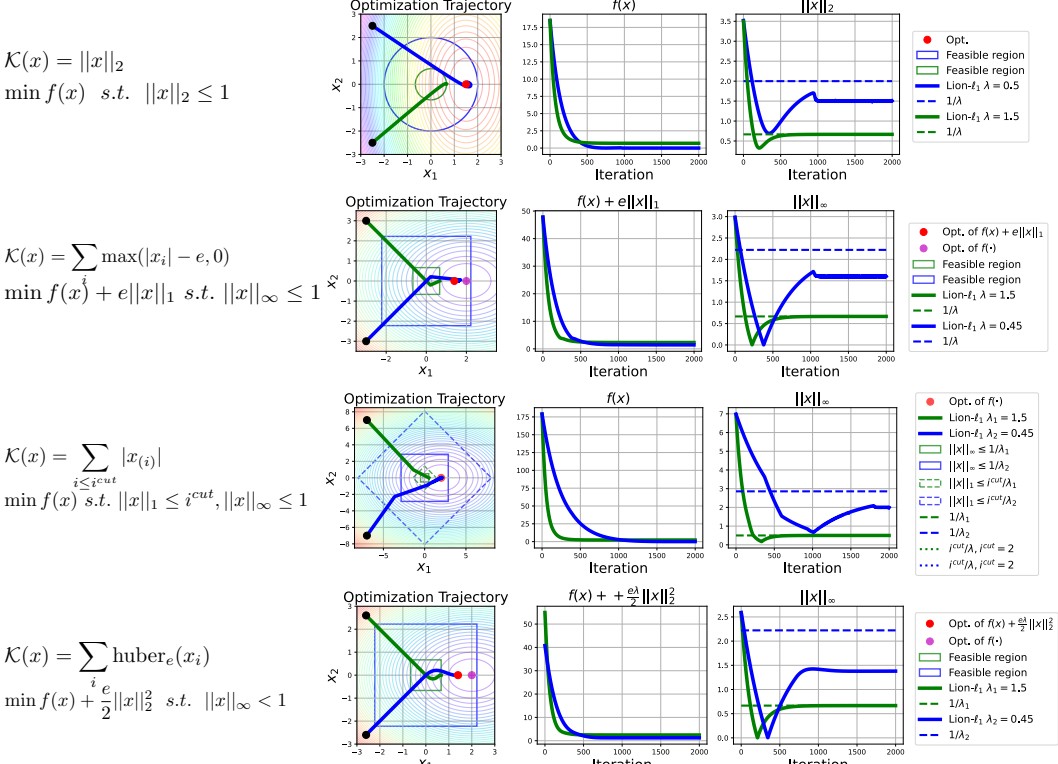

Figure 5: The behavior of Lion-$\mathcal{K}$ with different $\mathcal{K}$s from Table 2. The blue trajectory always reaches the optimum as the optimum is included in the constraint. The green trajectory converges to the boundary of the constraint.

**Experiment Setting** For the ImageNet training, we follow the standard PyTorch ImageNet training code.[1] We train the ResNet-50 and the ViT-B/16 model using batch size 1024 and cosine learning rate scheduler. For GPT-2 training, we follow the HuggingFace code[2], train it on OpenWebText[3] using cosine learning rate scheduler.

**Observation** From Figure 6, we observe that even on deep neural networks like ViT [9], ResNet [13], and GPT-2 [31], the behavior of the Lion-$\mathcal{K}$ optimizers strictly follow what the theory predicts. From Figure 7, we observe that Lion-$\ell_1$ (the original Lion optimizer) performs better than Lion with other $p$ on ImageNet when ViT is used, and on language modeling with the GPT-2 model. The plot indicates a trend that smaller $p \in [0, 1]$ results in better training efficiency. However, the trend is reversed when ResNet-50 [13] is used on ImageNet. Therefore, this indicates that the choice of $\mathcal{K}$ might depend on the underlying neural architecture. Based on the empirical observation, we conjecture that Lion-$\ell_1$ performs well among all Lion-$\ell_p$ on the transformer architecture, which is consistent with the fact that Lion-$\ell_1$ is found by an evolutionary search using the transformer architecture [6].

## 6 DISCUSSION

As demonstrated in the analysis of the Lyapunov function in Theorem 3.1, the Lion-$\mathcal{K}$ dynamics exhibit a distinct nature when compared to typical momentum-based methods like Polyak, Nesterov momentum, and Hamiltonian descent, all of which can be conveniently understood as certain generalized dissipative Hamiltonian systems. While the Lyapunov function provides a powerful

---

[1] https://github.com/pytorch/examples/blob/main/imagenet/main.py.
[2] https://huggingface.co/gpt2
[3] https://huggingface.co/datasets/Skylion007/openwebtext

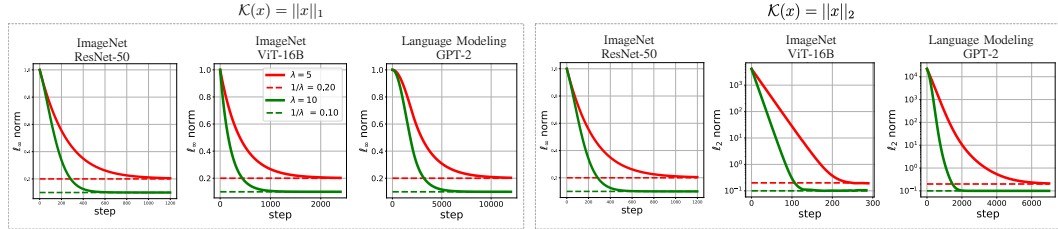

Figure 6: Constraint verification for Lion-$\ell_1$ and Lion-$\ell_2$ on ImageNet and Language Modeling tasks, using the ResNet-50, ViT-B/16 and the GPT-2 architectures.

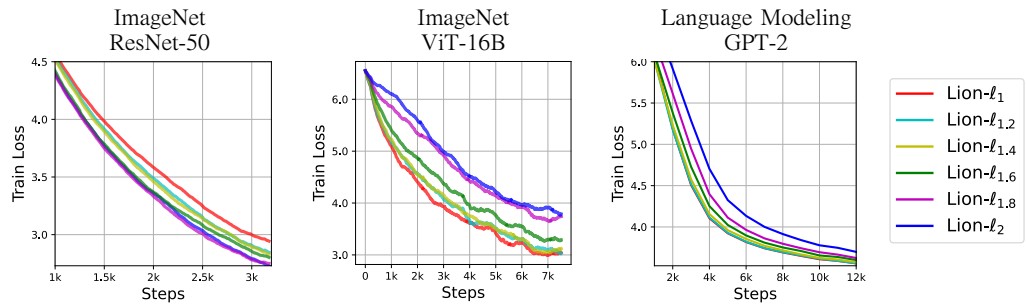

Figure 7: Performance of Lion-$\ell_p$ with different $p$, on ImageNet [34] (left 2 figures) and Language Modeling (right), using ResNet-50 [13] (left), ViT [9] (middle), and GPT-2 [31] (right).

characterization of the dynamical behavior, our intuitive understanding of the Lion-$\mathcal{K}$ dynamics remains obscured because we lack a "physical intuition" or constructive derivation like the standard optimization algorithms. This invites more studies in studies and understandings in future works.

The connection between Lion-$\mathcal{K}$ and Nesterov momentum and accelerated mirror descent suggests the possibility of acceleration phenomena in variants of Lion-$\mathcal{K}$, which opens an exciting avenue for future exploration and research. It might be possible to find novel accelerated algorithms based on the Lion-$\mathcal{K}$ family.

It is surprising and compelling that an algorithm found by a random search program has such a rich and intriguing theoretical basis. The reasons for this remain elusive, whether it is a coincidence or due to some inherent necessity. For instance, the design of the search space in Chen et al. [6] may in some way entails a high likelihood of discovering theoretically sound algorithms with random search. Understanding the underlying logic here could lead to future advancements in automatic machine-based algorithm discovery.

Regarding applications, since Lion-$\mathcal{K}$ offers a broader family than Lion, it is possible to find within the Lion-$\mathcal{K}$ family new algorithms that outperform Lion in various tasks and metrics. Additionally, by using different values of $\mathcal{K}$, Lion-$\mathcal{K}$ can be utilized to address different types of constraint optimization problems.

# 7 ACKNOWLEDGEMENT

The research is conducted in Statistics & AI group at UT Austin, which receives supports in part from NSF CAREER1846421, SenSE2037267, Office of Navy Research, and NSF AI Institute for Foundations of Machine Learning (IFML).

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

## A  EXAMPLES OF $\mathcal{K}$

We provide a list of examples of $\mathcal{K}$ and the corresponding $\nabla \mathcal{K}$ and $\mathcal{K}^*$. It is useful to define the following indicator functions of set $\{z = 0\}$:

$$\delta(z) = \begin{cases} 0 & \text{if } z = 0 \\ +\infty & \text{if } z \neq 0. \end{cases}, \qquad \mathbb{I}(z) = \begin{cases} 0 & \text{if } z = 0 \\ 1 & \text{if } z \neq 0. \end{cases},$$

Note that $\delta$ is the conjugate function of $f(x) = x$, as $\delta(x) = \sup_z x^\top z$.

$\ell_p$ **norm**  When $\mathcal{K}(x) = \|x\|_p = (\sum_i |x_i|^p)^{1/p}$ for $p \geq 1$, we can take

$$\nabla \mathcal{K}(x) = \frac{\text{sign}(x)\, |x|^{p-1}}{\|x\|_p^{p-1}},$$

and

$$\mathcal{K}^*(x) = \sup_z x^\top z - \|z\|_p = \sup_{c \geq 0} \|x\|_q\, c - c = \delta(\|x\|_q \leq 1),$$

where $q$ is the conjugate number of $p$, satisfying $\frac{1}{p} + \frac{1}{q} = 1$. Hence, Lion-$\mathcal{K}$ with $\ell_p$ norm correspond to solving

$$\min_x f(x) \quad s.t. \quad \|x\|_q \leq 1/\lambda.$$

**Group $\ell_p$ norm**  Assume $x$ is partitioned into a number of groups: $x = [x_{\mathcal{G}_i}]_{i=1}^k$. Consider the group $\ell_p$ norm: $\mathcal{K}(x) = \sum_{i=1}^k \|x_{\mathcal{G}_i}\|_p$. Then, we can take

$$\nabla \mathcal{K}(x) = \left[ \frac{\text{sign}(x_{\mathcal{G}_i})\, |x_{\mathcal{G}_i}|^{p-1}}{\|x_{\mathcal{G}_i}\|_p^{p-1}} \right]_{i=1}^k$$

The conjugate function is

$$\mathcal{K}^*(x) = \sup_z \sum_{i=1}^k x_{\mathcal{G}_i}^\top z_{\mathcal{G}_i} - \|z_{\mathcal{G}_i}\|_p = \sum_{i=1}^k \delta(\|x_{\mathcal{G}_i}\|_q \leq 1).$$

Hence, Lion-$\mathcal{K}$ with grouped $\ell_p$ norm corresponds to solving

$$\min_x f(x) \quad s.t. \quad \|x_{\mathcal{G}_i}\|_q \leq 1/\lambda, \quad \forall i.$$

**Lower Truncated $\ell_1$ Norm**  Consider $\mathcal{K}(x) = \sum_{i=1}^d \max(|x_i| - e, 0)$ where $e > 0$. We can take

$$\nabla \mathcal{K}(x) = \mathbb{I}(|x| \geq e)\text{sign}(x), \tag{16}$$

which uses $\text{sign}(x)$ as Lion, but zeros out the gradient on the elements with absolute values smaller than $e$. The conjugate is

$$\mathcal{K}^*(x) = \sup_z \sum_{i=1}^d (x_i z_i - \max(|z|_i - e, 0))$$

$$= \sup_{z,c} \sum_{i=1}^d (x_i z_i - c_i) \quad s.t. \quad c_i \geq 0, \quad c \geq |z_i| - e$$

$$= \sup_{c \geq 0} \sum_{i=1}^d |x_i|\,(c_i + e) - c_i$$

$$= \sum_{i=1}^d \delta(|x_i| \leq 1) + e\,|x_i|$$

$$= \delta(\|x\|_\infty \leq 1) + e\,\|x\|_1.$$

Hence, Lion-$\mathcal{K}$ corresponds to solving

$$\min_x \alpha f(x) + e\gamma \|x\|_1 \quad s.t. \quad \|x\|_\infty \leq 1/\lambda. \tag{17}$$

Hence, truncating the small gradients in Lion induces an $\ell_1$ penalty, which encourages the sparsity of the final solution.

**Lower (Vector-wise) Truncated $\ell_p$ Norm**    Consider $\mathcal{K}(x) = \max(\|x\|_p - e, 0)$. We have

$$\nabla \mathcal{K}(x) = \mathbb{I}(\|x\|_p - e \geq 0) \frac{\text{sign}(x) |x|^{p-1}}{\|x\|_p^{p-1}},$$

in which the gradient is zeroed out when $\|x\|_p \leq e$. The conjugate is

$$
\begin{aligned}
\mathcal{K}^*(x) &= \sup_z (x^\top z - \max(\|z\|_p - e, 0)) \\
&= \sup_{z,c} (x^\top z - c) \quad s.t. \quad c \geq 0, \quad c \geq \|z\|_p - e \\
&= \sup_{c \geq 0} \|x\|_q (c + e) - c \\
&= \delta(\|x\|_q \leq 1) + e \|x\|_q.
\end{aligned}
$$

Hence, Lion-$\mathcal{K}$ corresponds to solving

$$\min_x \alpha f(x) + e\gamma \|x\|_q \quad s.t. \quad \|x\|_q \leq 1/\lambda.$$

**Sorting Norm**    For $x = [x_1, \ldots, x_d]$, let $|x_{(1)}| \geq |x_{(2)}| \ldots$ be the sorting of the elements by absolute values. Define

$$\text{Sorting norm:} \quad \mathcal{K}(x) = \sum_i c_i |x_{(i)}|,$$

where $c_1 \geq c_2 \geq \ldots \geq 0$ is a descending non-negative sequence. The sorting norm is convex because it can be represented as the supreme of a set of convex functions, by the rearrangement inequality, as follows

$$\mathcal{K}(x) = \max_{\sigma \in \Gamma} \sum_{i=1}^{d} c_{\sigma(i)} |x_i|,$$

where $\Gamma$ denotes the set of permutations on $\{1, \ldots, n\}$. One subgradient of $\mathcal{K}$ is

$$\nabla \mathcal{K}(x)_i = c_{rank(i,x)} \text{sign}(x_i),$$

where $rank(i, x)$ denotes the rank of $|x_i|$ in $x$.

$$
\begin{aligned}
\mathcal{K}^*(x) &= \sup_z \left\{ x^\top z - \sum_i c_i |z_{(i)}| \right\} \\
&= \sup_{z \geq 0} \left\{ \sum_i |x_{(i)}| \times z_{(i)} - \sum_i c_i z_{(i)} \right\} && \text{//by rearrangement inequality} \\
&= \sup_{w \geq 0} \left\{ \sum_i (|x_{(i)}| - c_i) \times (\sum_{j \geq i} w_j) \right\} && \text{//let } z_{(i)} = \sum_{j \geq i} w_j, w_j \geq 0 \\
&= \sup_{w \geq 0} \left\{ \sum_j \sum_{i \leq j} (|x_{(i)}| - c_i) \times w_j \right\} && \text{//let } z_{(i)} = \sum_{j \geq i} w_j, w_j \geq 0 \\
&= \sum_j \delta(\sum_{i \leq j} |x_{(i)}| \leq \sum_{j \leq i} c_j)
\end{aligned}
$$

Hence, Lion-$\mathcal{K}$ corresponds to imposing a sequence of bounds on the cumsum of the sorted $x$:

$$\min_x f(x) \quad s.t. \quad \sum_{j \leq i} |x_{(i)}| \leq C_i, \quad \text{where } C_i = \sum_{j \leq i} c_j.$$

An interesting special case is when $c_i = \mathbb{I}(i \leq i^{cut})$ for some integer $i^{cut} \in \{1, \ldots, d\}$, so that

$$\mathcal{K}(x) = \sum_{i \leq i^{cut}} |x_{(i)}|, \qquad\qquad \nabla \mathcal{K}(x) = \mathbb{I}(|x| \geq x_{(i^{cut})}) \text{sign}(x),$$

in which we zero out the updates of the elements whose absolute values are smaller than the $i^{cut}$-th largest element. It is useful to compare this with (16) which applies the truncation based on a fixed number $\epsilon$, rather than the percentile.

The conjugate is

$$\mathcal{K}^*(x) = \sum_{j \leq i^{cut}} \delta(|x_{(j)}| \leq 1) + \delta(\|x\|_1 \leq i^{cut})$$

Then, Lion-$\mathcal{K}$ in this case corresponds to solving

$$\min_x f(x) \quad s.t. \quad \|x\|_1 \leq i^{cut}/\lambda, \quad \|x\|_\infty \leq 1/\lambda,$$

in which the percentile-based truncation effectively imposes a constraint on the $\ell_1$ norm of $x$. It is different from (17) in which the $\ell_1$ norm appears as a regularization term in the objective, rather than as a hard constraint.

**Entropy**   Consider $\mathcal{K}(x) = \sum_{i=1}^d \frac{1}{a} \log\left(\frac{1}{2}(\exp(ax_i) + \exp(-ax_i))\right)$, where $a > 0$. We have

$$\nabla \mathcal{K}(x) = \frac{\exp(ax) - \exp(-ax)}{\exp(ax) + \exp(-ax)} = \tanh(ax).$$

Taking the inverse, we have $\nabla \mathcal{K}^*(x) = \frac{1}{2a} \log \frac{1+x}{1-x}$, with domain in $\|x\|_\infty \leq 1$. by integration, the conjugate function is hence,

$$\mathcal{K}^*(x) = \sum_{i=1}^d \frac{1}{2a}(x_i + 1)\log(x_i + 1) + \frac{1}{2a}(1 - x_i)\log(1 - x_i) + \delta(\|x\|_\infty < 1).$$

Lion-$\mathcal{K}$ correspond to solving an entropy-regularized optimization:

$$\min_x \alpha f(x) + \frac{\gamma}{\lambda} E(\lambda x) \quad s.t. \quad \|x\|_\infty \leq 1/\lambda,$$

where $E(x) = \sum_{i=1}^d \frac{1}{2a}(x_i + 1)\log(x_i + 1)n + \frac{1}{2a}(1 - x_i)\log(1 - x_i)$.

**Huber Loss**   For $a \geq 0$, define the Huber loss:

$$\mathcal{K}(x) = \sum_{i=1}^d \text{Huber}_a(x_i) \quad \text{where} \quad \text{Huber}_a(x_i) = \mathbb{I}(|x_i| \geq a) \times |x_i| + \mathbb{I}(|x_i| < a) \times \frac{1}{2a}x_i^2,$$

We have

$$\nabla \mathcal{K}(x) = \text{Clip}(x, -a, a)/a, \quad \text{with} \quad \text{Clip}(x_i, a, b) = \begin{cases} x_i & \text{if } x \in [a, b] \\ b & \text{if } x > b \\ a & \text{if } x < a. \end{cases}$$

The conjugate is

$$\mathcal{K}^*(x) = \frac{a}{2}\|x\|_2^2 + \delta(\|x\|_\infty \leq 1),$$

$$\mathcal{K}^*(x) = \sum_{i=1}^d \max\left(\sup_{|z| \geq a} x_i z_i - |z_i|, \quad \sup_{|z_i| < a} x_i z_i - \frac{1}{2a}z_i^2\right)$$

$$= \sum_{i=1}^d \max\left(\delta(|x_i| \leq 1) + a(|x_i| - 1), \quad \frac{1}{2}ax_i^2\right)$$

$$= \sum_{i=1}^d \delta(|x_i| \leq 1) + \frac{1}{2}ax_i^2$$

$$= \frac{a}{2}\|x\|_2^2 + \delta(\|x\|_\infty \leq 1).$$

**Relativistic** Consider $\mathcal{K}(x) = \sum_{i=1}^{d} \sqrt{x_i^2 + e^2}$, then $\nabla \mathcal{K}(x) = \frac{x}{\sqrt{x^2 + e^2}}$, and

$$
\begin{aligned}
\mathcal{K}^*(x) &= \sup_z \left( \sum_{i=1}^{d} x_i z_i - \sqrt{z_i^2 + e^2} \right) \\
&= \sum_{i=1}^{d} \frac{x_i^2 e}{\sqrt{1 - x_i^2}} - \frac{e}{\sqrt{1 - x_i^2}} \qquad \text{//Solution: } z_i^2 = \frac{x_i^2 e^2}{1 - x_i^2} \\
&= \sum_{i=1}^{d} -e\sqrt{1 - x_i^2} + \delta(|x_i| \leq 1) \\
&= \sum_{i=1}^{d} -e\sqrt{1 - x_i^2} + \delta(\|x\|_\infty \leq 1).
\end{aligned}
$$

A related case is

$$
\mathcal{K}(x) = |x| - e\log(|x|/e + 1), \qquad \text{with} \qquad \nabla\mathcal{K}(x) = \frac{x}{|x| + e},
$$

whose conjugate function is

$$
\begin{aligned}
\mathcal{K}^*(x) &= \sup_x \left( \sum_{i=1}^{d} x_i z_i - |z_i| + e\log(|z_i|/e + 1) \right) \\
&= \sum_{i=1}^{d} |x_i|^2 e/(1 - |x_i|) - |x_i| e/(1 - |x_i|) + e\log(1/(1 - |x_i|)) \qquad \text{//Solution: } z = |x| e/(1 - |x|) \\
&= \sum_{i=1}^{d} -e(|x_i| + \log(1 - |x_i|)) + \delta(\|x\|_\infty < 1).
\end{aligned}
$$

## B  PROOFS

### B.1  CONVEX FUNCTION PRELIMINARIES

**Lemma 2.1**  *Assume $\mathcal{K}, \mathcal{K}^*$ is a closed convex conjugate pair and $\nabla\mathcal{K}$, $\nabla\mathcal{K}$ are their subgradients, we have*

$$
(\nabla\mathcal{K}(x) - \nabla\mathcal{K}(y))^\top (x - y) \geq 0, \qquad (\nabla\mathcal{K}(x) - y)^\top (x - \nabla\mathcal{K}^*(y)) \geq 0. \qquad (18)
$$

*Proof.* 1) By definition of subgradient, we have

$$
\begin{aligned}
\mathcal{K}(y) - \mathcal{K}(x) &\geq \nabla\mathcal{K}(x)^\top (y - x) \\
\mathcal{K}(x) - \mathcal{K}(y) &\geq \nabla\mathcal{K}(y)^\top (x - y).
\end{aligned}
$$

Summing them together yields $(\nabla\mathcal{K}(x) - \nabla\mathcal{K}(y))^\top (x - y) \geq 0$.

2) Because $\nabla\mathcal{K}^*(y) \in \partial\mathcal{K}^*(y)$, we have

$$
\mathcal{K}^*(\nabla\mathcal{K}(x)) - \mathcal{K}^*(y) \geq \nabla\mathcal{K}^*(y)^\top (\nabla\mathcal{K}(x) - y),
$$

Because $\nabla\mathcal{K}(x) \in \partial\mathcal{K}(x)$, by the property of conjugate functions, we have $x \in \partial\mathcal{K}^*(\nabla\mathcal{K}(x))$, and hence

$$
\mathcal{K}^*(y) - \mathcal{K}^*(\nabla\mathcal{K}(x)) \geq x^\top (y - \nabla\mathcal{K}(x)).
$$

Summing the two inequalities above yields

$$
(\nabla\mathcal{K}(x) - y)^\top (\nabla\mathcal{K}^*(y) - x) \leq (\mathcal{K}^*(\nabla\mathcal{K}(x)) - \mathcal{K}^*(y)) + (\mathcal{K}^*(y) - \mathcal{K}^*(\nabla\mathcal{K}(x))) = 0.
$$

$\square$

**Lemma B.1.** *The Lion-$\mathcal{K}$ ODE is*

$$\dot{x}_t = \nabla\mathcal{K}(m_t - \varepsilon(\alpha\nabla f(x_t) + \gamma m_t)) - \lambda x_t$$
$$\dot{m}_t = -\alpha\nabla f(x_t) - \gamma m_t.$$

*is equivalent to*

$$\nabla^2\mathcal{K}^*(\dot{x}_t + \lambda x_t)(\ddot{x}_t + \lambda\dot{x}_t) + \varepsilon\alpha\nabla^2 f(x_t)\dot{x}_t + \gamma\nabla\mathcal{K}^*(\dot{x}_t + \lambda x_t) + \alpha\nabla f(x_t) = 0, \qquad (19)$$

*if $\mathcal{K}^*$ and $f$ are second order differentiable.*

*In particular, if $\mathcal{K}(x) = \|x\|_2^2/2$, we have*

$$\ddot{x}_t + (\lambda + \gamma)\dot{x}_t + \varepsilon\alpha\nabla^2 f(x_t)\dot{x}_t + \gamma\lambda x_t + \alpha\nabla f(x_t) = 0. \qquad (20)$$

*This ODE minimizes $F(x) = \alpha f(x) + \gamma\lambda\|x\|_2^2/2$.*

**Remark**  We have the following observations from (21):

1) The role of the weight decay $\lambda$ and momentum damping coefficient $\gamma$ is symmetric in (21).

2) When either the weight decay or momentum damping is turned off, i.e., $\gamma\lambda = 0$, the $\ell_2$ regularization in $F(x)$ is turned off, and we have

$$\ddot{x}_t + (\lambda + \gamma)\dot{x}_t + \varepsilon\alpha\nabla^2 f(x_t)\dot{x}_t + \alpha\nabla f(x_t) = 0, \qquad (21)$$

which coincides with the *high-resolution* ODE [35] that serves as a continuous-time modeling of Nesterov momentum for minimizing $f(x)$.

3) The Hessian-dependent damping term $\nabla^2 f(x_t)\dot{x}_t$ arises to due the gradient enhancement ($\varepsilon > 0$), and it is known to play a key role in Nesterov momentum and acceleration [1, 35]. When we turn off the gradient enhancement ($\varepsilon = 0$), we get

$$\ddot{x}_t + (\lambda + \gamma)\dot{x}_t + \alpha\nabla f(x_t) = 0,$$

which is the ODE for Polayk momentum, the equation of motion of a ball with unit mass moving in a potential field $\alpha f(x)$ with a friction coefficient $(\lambda + \gamma)$.

*Proof.* We want to cancel out $m_t$. The first equation yields

$$(1 - \varepsilon\gamma)m_t = (\nabla\mathcal{K}^*(\dot{x}_t + \lambda x_t) + \varepsilon\alpha\nabla f(x_t)). \qquad (22)$$

Plugging it into the second equation yields

$$(1 - \varepsilon\gamma)\dot{m}_t = -\alpha(1 - \varepsilon\gamma)\nabla f(x_t) - \gamma(\nabla\mathcal{K}^*(\dot{x}_t + \lambda x_t) + \varepsilon\alpha\nabla f(x_t))$$
$$= -\alpha\nabla f(x_t) - \gamma\nabla\mathcal{K}^*(\dot{x}_t + \lambda x_t). \qquad (23)$$

Combining (22) and (23) yields

$$\frac{\mathrm{d}}{\mathrm{d}t}(\nabla\mathcal{K}^*(\dot{x}_t + \lambda x_t) + \varepsilon\alpha\nabla f(x_t)) = -\alpha\nabla f(x_t) - \gamma\nabla\mathcal{K}^*(\dot{x}_t + \lambda x_t).$$

Or

$$\nabla^2\mathcal{K}^*(\dot{x}_t + \lambda x_t)(\ddot{x}_t + \lambda\dot{x}_t) + \varepsilon\alpha\nabla^2 f(x_t)\dot{x}_t + \gamma\nabla\mathcal{K}^*(\dot{x}_t + \lambda x_t) + \alpha\nabla f(x_t) = 0.$$

$\square$

## B.3 DISCRETE-TIME SCHEMES OF LION-$\mathcal{K}$

In the most general form, the Euler approximation of the Lion-$\mathcal{K}$ ODE with step size $\epsilon$ is

$$x_{t+1} = x_t + \epsilon(\nabla\mathcal{K}(m_t - \varepsilon(\alpha\nabla f(x_t) + \gamma m_t)) - \lambda x_t)$$
$$m_{t+1} = m_t - \epsilon(\alpha\nabla f(x_t) + \gamma m_t), \qquad (24)$$

The discrete Lion-$\mathcal{K}$ scheme in (2) is recovered when $\alpha = \gamma$, $\beta_1 = 1 - \varepsilon\gamma$, $\beta_2 = 1 - \epsilon\gamma$. By scaling $f(x)$ by a positive multiplicative ratio, (2) in fact covers all cases of (24) when $\gamma \neq 0$.

When $\gamma = 0$, however, (24) reduces to a momentum-undamped variant of Lion-$\mathcal{K}$:

$$\text{Undamped Lion-}\mathcal{K}: \quad \begin{aligned} x_{t+1} &= x_t + \epsilon(\nabla\mathcal{K}(m_t - \beta_1\nabla f(x_t)) - \lambda x_t) \\ m_{t+1} &= m_t - \beta_2\nabla f(x_t), \end{aligned}$$

which is the Euler approximation of Lion-$\mathcal{K}$ ODE $\gamma = 0$, step size $\epsilon$, and $\beta_1 = \varepsilon\alpha$, and $\beta_2 = \epsilon\alpha$. Due to $\gamma = 0$, the undamped Lion-$\mathcal{K}$ amounts to solving $\min_x f(x)$, without the regularization $\mathcal{K}^*(\lambda x)$.

The connection to Polyak and Nesterov momentum discussed in Section extends to discrete-time forms. From the first equation (24), we have

$$m_t = \frac{1}{1 - \varepsilon\gamma}\left(\nabla\mathcal{K}^*\left(\frac{x_{t+1} - x_t}{\epsilon} + \lambda x_t\right) + \varepsilon\alpha\nabla f(x_t)\right).$$

Plugging it into the second equation of (24), we get

$$\left(\nabla\mathcal{K}^*\left(\frac{x_{t+2} - x_{t+1}}{\epsilon} + \lambda x_{t+1}\right) + \varepsilon\alpha\nabla f(x_{t+1})\right) = (1 - \epsilon\gamma)\left(\nabla\mathcal{K}^*\left(\frac{x_{t+1} - x_t}{\epsilon} + \lambda x_t\right) + \varepsilon\alpha\nabla f(x_t)\right) - (1 - \varepsilon\gamma)\epsilon\alpha\nabla f(x_t).$$

Hence,

$$\nabla\mathcal{K}^*\left(\frac{x_{t+2} - x_{t+1}}{\epsilon} + \lambda x_{t+1}\right) = -\varepsilon\alpha\nabla f(x_{t+1}) + (1 - \epsilon\gamma)\nabla\mathcal{K}^*\left(\frac{x_{t+1} - x_t}{\epsilon} + \lambda x_t\right) + (\varepsilon - \epsilon)\alpha\nabla f(x_t).$$

When $\nabla\mathcal{K}^*(x) = x$, we have

$$x_{t+2} = (1 - \epsilon\lambda)x_{t+1} - \epsilon\varepsilon\alpha\nabla f(x_{t+1}) + (1 - \epsilon\gamma)((x_{t+1} - x_t) + \epsilon\lambda x_t) + \epsilon(\varepsilon - \epsilon)\alpha\nabla f(x_t).$$

It is simplified into

$$x_{t+2} = (1 - \epsilon^2\lambda\gamma)x_{t+1} - \epsilon^2\alpha\nabla f(x_{t+1}) + (1 - \epsilon\gamma)(1 - \epsilon\lambda)(x_{t+1} - x_t) - \epsilon(\varepsilon - \epsilon)\alpha(\nabla f(x_{t+1}) - \nabla f(x_t)).$$

When $\varepsilon > \epsilon$ (corresponding to $\beta_1 < \beta_2$ in Lion-$\mathcal{K}$ (2)), this can be shown to be identical to the Nesterov momentum algorithm for minimizing $F(x) = \alpha f(x) + \lambda\gamma\|x\|_2^2/2$. When $\varepsilon = \epsilon$ (corresponding to $\beta_1 = \beta_2$ in (2)), it is identical to Polyak momentum.

### B.4   FRANK-WOLFE AND MIRROR DESCENT

**Frank-Wolfe**   When $\varepsilon\gamma = 1$, Lion-$\mathcal{K}$ reduces to

$$\dot{x}_t = \nabla\mathcal{K}(-\nabla f(x_t)) - \lambda x_t, \tag{25}$$

where we also set $\varepsilon\alpha = 1$ without loss of generality. In this case, the ODE monotonically decreases the objective

$$F(x) = f(x) + \frac{1}{\lambda}\mathcal{K}^*(\lambda x),$$

without resorting to an additional Lyapunov function. This can be seen from

$$\frac{\mathrm{d}}{\mathrm{d}t}F(x_t) = (\nabla f(x) + \nabla\mathcal{K}^*(\lambda x))^\top(\nabla\mathcal{K}(-\nabla f(x)) - \lambda x) \leq 0,$$

where the inequality follows Lemma 2.1.

The Euler discretization of (25) is

$$x_{t+1} = x_t + \epsilon\left(\nabla\mathcal{K}(-\nabla f(x_t)) - \lambda x_t\right). \tag{26}$$

This can also be derived from conditional gradient descent, or Frank–Wolfe. To see this, recall that the conditional gradient descent update for the $F(x)$ above is

$$\begin{aligned} y_{t+1} &= \arg\min_x\left\{\nabla f(x_t)^\top(x - x_t) + \frac{1}{\lambda}\mathcal{K}^*(\lambda x)\right\} \\ x_{t+1} &= x_t + \epsilon_0(y_{t+1} - x_t), \end{aligned}$$

Solving $y_{t+1}$ yields

$$y_{t+1} = \frac{1}{\lambda}\nabla\mathcal{K}(-\nabla f(x_t)), \qquad \text{and hence} \qquad x_{t+1} = (1 - \epsilon_0)x_t + \frac{\epsilon_0}{\lambda}\nabla\mathcal{K}(-\nabla f(x_t)).$$

Taking $\epsilon = \epsilon_0\lambda$ yields (26).

**Dual Space Preconditioning and Mirror Descent**   When we further set $\lambda = 0$ in (26), Lion-$\mathcal{K}$ reduces to

$$x_{t+1} = x_t + \epsilon \nabla \mathcal{K}(-\nabla f(x_t)), \tag{27}$$

When $\nabla \mathcal{K}(0) = 0$, Eq. (27) is dual space preconditioning [23], which is closely related to mirror descent [26], for minimizing $f(x)$. To see the connection with mirror descent, note that (27) is equivalent to

$$x_{t+1} = x_t + \epsilon \delta_t, \qquad \text{with} \qquad \delta_t = \arg\min_\delta \left\{ \nabla f(x_t)^\top \delta + \mathcal{K}^*(\delta) \right\}.$$

Because $\mathcal{K}^*$ and $\mathcal{K}$ are differentiable, then $\nabla \mathcal{K}(0) = 0$ implies $\nabla \mathcal{K}^*(0) = 0$, and hence $\mathcal{K}^*$ achieves the minimum at zero. In this case, $\mathcal{K}^*(\delta) - \mathcal{K}^*(0)$ can be viewed as a Bregman divergence, and hence justifying the connection of (27) with mirror descent. Recall that the Bregman divergence $B_h(x \,\|\, y)$ is the Bregman divergence associated with a convex function $h \colon \mathbb{R}^d \to \mathbb{R}$ is defined as

$$B_h(x \,\|\, y) = h(x) - h(y) - \nabla h(y)^\top (x - y).$$

With $\nabla \mathcal{K}^*(0) = 0$, it is then easy to show

$$\mathcal{K}^*(\delta) - \mathcal{K}^*(0) = B_{\mathcal{K}^*}(\delta \,\|\, 0) = B_{\mathcal{K}_t^*}(x_t + \epsilon \delta \,\|\, x_t),$$

where $\mathcal{K}_t^* = \mathcal{K}^*\left(\frac{x - x_t}{\epsilon}\right)$.

## B.5   LION-$\mathcal{K}$ WITHOUT GRADIENT ENHANCEMENT ($\varepsilon = 0$)

**Theorem B.2.** *Consider the ODE of Lion-$\mathcal{K}$-W without gradient correction:*

$$\begin{aligned} \dot{x}_t &= \nabla \mathcal{K}(m_t) - \lambda x_t \\ \dot{m}_t &= -\alpha \nabla f(x_t) - \gamma m_t, \end{aligned} \tag{28}$$

*with $\lambda, \alpha, \gamma > 0$. Its fixed point is the minimum of*

$$\min_x \alpha f(x) + \frac{\gamma}{\lambda} \mathcal{K}^*(\lambda x).$$

*It yields the following Lyapunov function:*

$$H(x, m) = \alpha f(x) + \frac{\gamma}{\lambda} \mathcal{K}^*(\lambda x) + (\mathcal{K}^*(\lambda x) + \mathcal{K}(m) - \lambda x^\top m).$$

*Proof.*   Observe that

$$\begin{aligned} \nabla_x H(x, m) &= \alpha \nabla f(x) + (\gamma + \lambda)\nabla \mathcal{K}^*(\lambda x) - \lambda m \\ \nabla_m H(x, m) &= \nabla \mathcal{K}(m) - \lambda x, \end{aligned}$$

and (28) can be written into

$$\begin{aligned} \dot{x}_t &\coloneqq V_x(x_t, m_t) = \nabla_m H(x_t, m_t) \\ \dot{m}_t &\coloneqq V_m(x_t, m_t) = -\nabla_x H(x_t, m_t) - \hat{H}_m(x_t, m_t), \end{aligned}$$

with $\hat{H}_m(x_t, m_t) = (\gamma + \lambda)(m_t - \nabla \mathcal{K}^*(\lambda x_t))$. By Lemma 2.1, we have

$$\hat{H}_m^\top(\nabla_m H) = (m - \nabla \mathcal{K}^*(\lambda x))^\top (\nabla \mathcal{K}(m) - \lambda x) \geq 0.$$

Then

$$\begin{aligned} \frac{\mathrm{d}}{\mathrm{d}t} H(x_t, m_t) &= \nabla_x H^\top V_x + \nabla_m H^\top V_m \\ &= \nabla_x H^\top (\nabla_m H) + \nabla_m H^\top (-\nabla_x H - \hat{H}_m) = -\nabla_m H^\top \hat{H}_m \leq 0. \end{aligned}$$

In fact, this ODE has a Hamiltonian + descent structure [22], as it can viewed as a Hamiltonian system damped with a descending force:

$$\begin{bmatrix} \dot{x}_t \\ \dot{m}_t \end{bmatrix} = \underbrace{\begin{bmatrix} +\nabla_m H(x_t, m_t) \\ -\nabla_x H(x_t, m_t) \end{bmatrix}}_{\text{Hamiltonian}} - \underbrace{\begin{bmatrix} 0 \\ (\gamma + \lambda)(m_t - \nabla \mathcal{K}^*(\lambda x_t)) \end{bmatrix}}_{\text{Descent}},$$

where the Hamiltonian component is orthogonal to the gradient $[\nabla_x H, \nabla_m H]$ of $H(x, m)$ and preserves the total energy $H(x, m)$, and the descent component introduces a damping like effect to decrease the energy $H(x, m)$. □

### B.6 Lion-$\mathcal{K}$ without Weight Decay – A Hamiltonian + Descent Derivation

When the weight decay in Lion-$\mathcal{K}$ is turned off ($\lambda = 0$), there is an alternative way to analyze it that is amendable to the Hamiltonian + descent structure in (12).

Recall that the Lion-$\mathcal{K}$ ODE is of the following form when $\lambda = 0$:

$$\dot{x}_t = \nabla\mathcal{K}(\tilde{m}_t), \qquad \tilde{m}_t = m_t - \varepsilon(\alpha\nabla f(x_t) + \gamma m_t)$$
$$\dot{m}_t = -\alpha\nabla f(x_t) - \gamma m_t \tag{29}$$

Assume $\varepsilon\gamma < 1$. Define $\tilde{\mathcal{K}}(m) = \frac{1}{1-\varepsilon\gamma}\mathcal{K}((1-\varepsilon\gamma)m)$, and the following Lyapunov function:

$$H(x,m) = \alpha f(x) + \tilde{\mathcal{K}}(m) = \alpha f(x) + \frac{1}{1-\varepsilon\gamma}\mathcal{K}((1-\varepsilon\gamma)m). \tag{30}$$

Note that $\nabla_x H(x,m) = \alpha\nabla f(x)$ and $\nabla_m H(x,m) = \nabla\mathcal{K}((1-\varepsilon)m)$. One can decompose (29) into the following Hamiltonian + descent decomposition:

$$\begin{bmatrix} \dot{x}_t \\ \dot{m}_t \end{bmatrix} = \underbrace{\begin{bmatrix} +\nabla_m H(x_t, m_t) \\ -\nabla_x H(x_t, m_t) \end{bmatrix}}_{\text{Hamiltonian}} - \underbrace{\begin{bmatrix} \nabla\mathcal{K}(\tilde{m}_t^0) - \nabla\mathcal{K}(\tilde{m}_t) \\ \gamma m_t \end{bmatrix}}_{\text{Descent}},$$

where we define $\tilde{m}_t^0 = (1-\varepsilon\gamma)m_t$ and hence $\tilde{m}_t - \tilde{m}_t^0 = -\varepsilon\alpha\nabla f(x_t)$.

Using the monotonicity of subgradient (Lemma 2.1), one can show that the second component in the decomposition above is a descent direction of $H(x,m)$ in (30):

1) Let $\hat{\nabla}_x H_t := -\nabla\mathcal{K}(\tilde{m}_t^0) + \nabla\mathcal{K}(\tilde{m}_t)$, then it is a descent direction of $H(x,m)$, because

$$\nabla_x H(x_t, m_t)^\top \hat{\nabla}_x H_t = \alpha\nabla f(x_t)^\top \hat{\nabla}_x H_t$$
$$= -\frac{1}{\varepsilon}(\tilde{m}_t^0 - \tilde{m}_t)^\top(\nabla\mathcal{K}(\tilde{m}_t^0) - \nabla\mathcal{K}(\tilde{m}_t)) \leq 0,$$

where we used the monotonicity of $\nabla\mathcal{K}(\cdot)$.

2) If $m = 0$ is the minimum of $\mathcal{K}$, then $\hat{\nabla}_m H_t := -\gamma m_t$ is a descent direction of $H(x,m)$ because,

$$\nabla_m H(x_t, m_t)^\top \hat{\nabla}_m H_t = -\gamma\nabla\mathcal{K}((1-\varepsilon\gamma)m_t)^\top m_t \leq \frac{\gamma}{1-\varepsilon\gamma}(\mathcal{K}(0) - \mathcal{K}((1-\varepsilon\gamma)m_t)) \leq 0.$$

Hence, we have

$$\frac{\mathrm{d}}{\mathrm{d}t}H(x_t, m_t) = \nabla_x H(x_t, m_t)^\top \hat{\nabla}_x H_t + \nabla_m H(x_t, m_t)^\top \hat{\nabla}_m H_t$$
$$= -\frac{1}{\varepsilon}(\tilde{m}_t^0 - \tilde{m}_t)^\top(\nabla\mathcal{K}(\tilde{m}_t^0) - \nabla\mathcal{K}(\tilde{m}_t)) - \gamma\nabla\mathcal{K}((1-\varepsilon\gamma)m_t)^\top m_t \leq 0.$$

Moreover, if $m = 0$ is the unique minimum of $\mathcal{K}$, and $\varepsilon\gamma < 1$, then $\nabla\mathcal{K}((1-\varepsilon\gamma)m_t)^\top m_t = 0$ implies that $m_t = 0$, and one can show that the equilibrium points of (29) are stationary points of $H(x,m)$ using LaSalle's invariance principle.

### B.7 Main Result of Lion-$\mathcal{K}$ ODE

**Theorem B.3.** *Assume $\mathcal{K}$ is convex with conjugate $\mathcal{K}^*$. Assume $f, \mathcal{K}, \mathcal{K}^*$ are continuously differentiable. Assume $(x_t, m_t)$ is the solution of the following ODE:*

$$\dot{x}_t = \nabla\mathcal{K}(\tilde{m}_t) - \lambda x_t, \qquad \text{with} \qquad \tilde{m}_t = m_t - \varepsilon(\gamma m_t + \alpha\nabla f(x_t)),$$
$$\dot{m}_t = -\alpha\nabla f(x_t) - \gamma m_t,$$

*where $\alpha, \gamma, \lambda, \varepsilon > 0$ and $\epsilon\gamma \leq 1$. Let*

$$H(x,m) = \alpha f(x) + \frac{\gamma}{\lambda}\mathcal{K}^*(\lambda x) + \frac{1-\varepsilon\gamma}{1+\varepsilon\lambda}(\mathcal{K}^*(\lambda x) + \mathcal{K}(m) - \lambda m^\top x).$$

*Then H yields a Lyapunov function in that*

$$-\frac{\mathrm{d}}{\mathrm{d}t}H(x_t, m_t) = \Delta(x_t, m_t) := \frac{\lambda + \gamma}{1 + \varepsilon\lambda}\Delta_1(x_t, \tilde{m}_t) + \frac{1 - \varepsilon\gamma}{(1 + \varepsilon\lambda)}\Delta_2(m_t, \tilde{m}_t) \geq 0,$$

*where*

$$\Delta_1(x, \tilde{m}) = (\tilde{m} - \nabla\mathcal{K}^*(\lambda x))^\top(\nabla\mathcal{K}(\tilde{m}) - \lambda x),$$

$$\Delta_2(m, \tilde{m}) = \frac{1}{\varepsilon}(\tilde{m} - m)^\top(\nabla\mathcal{K}(\tilde{m}) - \nabla\mathcal{K}(m)).$$

*Moreover, the accumulation points of all trajectories are stationary points of $F(x) = \alpha f(x) + \frac{\gamma}{\lambda}\mathcal{K}^*(\lambda x)$.*

*Proof.* It is not obvious how to construct the Lyapunov function directly from the ODE. The following proof describes the process of discovering $H(x, m)$. We start by examing what inequalities we can write down using the monotonicity of $\nabla\mathcal{K}$ and $\nabla\mathcal{K}^*$ via Lemma 2.1, and then work out the Lyapunov function backward.

Write $\tilde{m} = m - \varepsilon(\gamma m + \alpha\nabla f(x))$. Because $\nabla\mathcal{K}$ is a monotonic mapping, we have by Lemma 2.1 the following key inequalities:

$$(-\tilde{m} + \nabla\mathcal{K}^*(\lambda x))^\top(\nabla\mathcal{K}(\tilde{m}) - \lambda x) \leq 0,$$

$$(m - \tilde{m})^\top(\nabla\mathcal{K}(\tilde{m}) - \nabla\mathcal{K}(m)) \leq 0,$$

or equivalently

$$(\varepsilon\alpha\nabla f(x) - (1 - \varepsilon\gamma)m + \nabla\mathcal{K}^*(\lambda x))^\top(\nabla\mathcal{K}(\tilde{m}) - \lambda x) \leq 0 \tag{31}$$

$$\varepsilon(\alpha\nabla f(x) + \gamma m)^\top((\nabla\mathcal{K}(\tilde{m}) - \lambda x) - (\nabla\mathcal{K}(m) - \lambda x)) \leq 0 \tag{32}$$

Write $V_x = \nabla\mathcal{K}(\tilde{m}) - \lambda x$, and $V_m = -\alpha\nabla f(x) - \gamma m$. So the ODE is $\dot{x} = V_x$ and $\dot{m} = V_m$. The inequalities can be rewritten into

$$(\varepsilon\alpha\nabla f(x) - (1 - \varepsilon\gamma)m + \nabla\mathcal{K}^*(\lambda x))^\top V_x \leq 0 \tag{33}$$

$$-\varepsilon V_m^\top(V_x - (\nabla\mathcal{K}(m) - \lambda x)) \leq 0 \tag{34}$$

Taking $\frac{1}{\varepsilon(1+\eta)}(Eq.\ (33) + \eta \times Eq.\ (34))$ for any $\eta \geq 0$, we get

$$\left(\alpha\nabla f(x) - \frac{1 - \varepsilon\gamma(1 + \eta)}{\varepsilon(1 + \eta)}m + \frac{1}{\varepsilon(1 + \eta)}\nabla\mathcal{K}^*(\lambda x)\right)^\top V_x + \frac{\eta\varepsilon}{\varepsilon(1 + \eta)}(\nabla\mathcal{K}(m) - \lambda x)^\top V_m \leq 0$$

Define

$$\tilde{H}(x, m) = \alpha f(x) + \frac{1}{\varepsilon(1 + \eta)\lambda}\mathcal{K}^*(\lambda x) + \frac{1 - \varepsilon\gamma(1 + \eta)}{\varepsilon(1 + \eta)}\frac{1}{\lambda}\mathcal{K}(m) - \frac{1 - \varepsilon\gamma(1 + \eta)}{\varepsilon(1 + \eta)}m^\top x.$$

Then the inequality was reduced to

$$\nabla_x\tilde{H}(x, m)^\top V_x + \frac{\varepsilon\eta\lambda}{1 - \varepsilon\gamma(1 + \eta)}\nabla_m\tilde{H}(x, m)^\top V_m \leq 0.$$

If we take $\eta$ such that

$$\frac{\varepsilon\eta\lambda}{1 - \varepsilon\gamma(1 + \eta)} = 1, \tag{35}$$

then we have when following $\dot{x} = V_x$ and $\dot{m} = V_m$,

$$\frac{\mathrm{d}}{\mathrm{d}t}\tilde{H}(x, m) = \nabla_x\tilde{H}(x, m)^\top V_x + \nabla_m\tilde{H}(x, m)^\top V_m \leq 0.$$

Furthermore, when (35) holds, we have

$$\eta = \frac{1 - \varepsilon\gamma}{\varepsilon(\lambda + \gamma)}, \qquad \frac{1}{\varepsilon(1 + \eta)} = \frac{\lambda + \gamma}{1 + \varepsilon\lambda}, \qquad \frac{1 - \varepsilon\gamma(1 + \eta)}{\varepsilon(1 + \eta)} = \frac{(1 - \varepsilon\gamma)\lambda}{1 + \varepsilon\lambda}, \tag{36}$$

and hence

$$\tilde{H}(x,m) = \alpha f(x) + \left(\frac{\lambda+\gamma}{(1+\epsilon\lambda)\lambda} - \frac{1-\varepsilon\gamma}{1+\varepsilon\lambda}\right)\mathcal{K}^*(\lambda x) + \frac{1-\varepsilon\gamma}{1+\varepsilon\lambda}\left(\mathcal{K}^*(\lambda x) + \mathcal{K}(m) - \lambda m^\top x\right)$$

$$= \alpha f(x) + \frac{\gamma}{\lambda}\mathcal{K}^*(\lambda x) + \frac{1-\varepsilon\gamma}{1+\varepsilon\lambda}\left(\mathcal{K}^*(\lambda x) + \mathcal{K}(m) - \lambda m^\top x\right)$$

$$= H(x,m).$$

In this case,

$$\frac{\mathrm{d}}{\mathrm{d}t}H(x,m)$$

$$= \frac{1}{\varepsilon(1+\eta)}\left(Eq.~(33) + \eta \times Eq.~(34)\right)$$

$$= \frac{\lambda+\gamma}{1+\varepsilon\lambda} \times Eq.~(33) + \frac{1-\varepsilon\gamma}{\varepsilon(1+\varepsilon\lambda)} \times Eq.~(34). \qquad \textcolor{magenta}{//\frac{\eta}{\varepsilon(1+\eta)} = \frac{1-\varepsilon\gamma}{\varepsilon(1+\varepsilon\lambda)} \text{ from } (36)}$$

$$= -\frac{\lambda+\gamma}{1+\varepsilon\lambda}(\tilde{m} - \nabla\mathcal{K}^*(\lambda x))^\top(\nabla\mathcal{K}(\tilde{m}) - \lambda x) - \frac{1-\epsilon\gamma}{(1+\epsilon\lambda)\varepsilon}(\tilde{m} - m)^\top(\nabla\mathcal{K}(\tilde{m}) - \nabla\mathcal{K}(m)) \le 0.$$

To ensure that $\eta \ge 0$, we need $\varepsilon\gamma \le 1$. $\qquad\qquad\square$

**LaSalle's invariance principle**  Let $H(z)$ is a continuously differentiable Lyapunov function of $\frac{\mathrm{d}}{\mathrm{d}t}z_t = v(z_t)$, satisfying $\frac{\mathrm{d}}{\mathrm{d}t}H(z_t) \le 0$. By LaSalle's Invariance Principle, the accumulation points of any trajectories of $\frac{\mathrm{d}}{\mathrm{d}t}z_t = v(z_t)$ is included in

$$\mathcal{I} = \{\text{the union of all trajectories } z_t \text{ satisfying } \frac{\mathrm{d}}{\mathrm{d}t}H(z_t) = 0 \text{ for all } t \ge 0\,\}.$$

For the Lion-$\mathcal{K}$ ODE and its $H$, the points in $\mathcal{I}$ should satisfy $\tilde{m}_t = \nabla\mathcal{K}^*(\lambda x_t)$, which yields $\nabla\mathcal{K}(\tilde{m}_t) = \lambda x_t$, and hence

$$\dot{x}_t = \nabla\mathcal{K}(\tilde{m}_t) - \lambda x_t = 0.$$

This suggests that $x_t$ is constant for the trajectories in $\mathcal{I}$. Because $\tilde{m}_t = \nabla\mathcal{K}^*(\lambda x_t)$ and $\tilde{m}_t = m_t - \varepsilon(\alpha\nabla f(x_t) + \gamma m_t)$, we have

$$(1-\varepsilon\gamma)m_t = \nabla\mathcal{K}^*(\lambda x_t) + \varepsilon\alpha\nabla f(x_t)$$

Hence, $(1-\varepsilon\gamma)m_t$ is also constants in the trajectories in $\mathcal{I}$. This suggests that $(1-\varepsilon\gamma)\dot{m}_t = 0$ along the trajectories in $\mathcal{I}$, and hence

$$0 = (1-\varepsilon\gamma)\dot{m}_t$$

$$= -(1-\varepsilon\gamma)(\alpha\nabla f(x_t) + \gamma m_t)$$

$$= -(1-\varepsilon\gamma)\alpha\nabla f(x_t) - \gamma\nabla\mathcal{K}^*(\lambda x_t) - \varepsilon\gamma\alpha\nabla f(x_t)$$

$$= -\alpha\nabla f(x_t) - \gamma\nabla\mathcal{K}^*(\lambda x_t)$$

$$= -\nabla F(x_t) \qquad \textcolor{magenta}{//F(x) = \alpha f(x) + \frac{\gamma}{\lambda}\mathcal{K}^*(\lambda x)}$$

Hence, all trajectories in $\mathcal{I}$ are singleton points and are stationary points of the objective $F(x) = \alpha f(x) + \frac{\gamma}{\lambda}\mathcal{K}^*(\lambda x)$.

## B.8   THE DECOMPOSITION STRUCTURE

We provide the decomposition structure (11) which provides a simplified proof of the Lyapunov property.

**Lemma B.4.**  *For ODE $\dot{x}_t = V_x(x_t, m_t)$, $\dot{m}_t = V_m(x_t, m_t)$, let $H(x,m)$ be a function satisfying*

$$\nabla_x H(x,m) = -\tilde{V}_x(x,m) + \eta V_m(x,m)$$

$$\nabla_m H(x,m) = -\hat{V}_m(x,m) - \eta V_x(x,m),$$

where $a \in \mathbb{R}$ and $\hat{V}_x$ and $\hat{V}_m$ have positive inner products with $V_x$, $V_m$, respectively, that is,

$$\hat{V}_x(x,m)^\top V_x(x,m) \geq 0, \qquad \hat{V}_m(x,m)^\top V_m(x,m) \geq 0, \quad \forall x, m.$$

Then we have

$$\frac{\mathrm{d}}{\mathrm{d}t} H(x_t, m_t) \leq 0.$$

*Proof.*

$$
\begin{aligned}
\frac{\mathrm{d}}{\mathrm{d}t} H(x_t, m_t) &= \nabla_x H^\top V_x + \nabla_m H^\top V_m \\
&= (-\hat{V}_x + aV_m)^\top V_x + (-\hat{V}_m - aV_x)^\top V_m \\
&= -(\hat{V}_x^\top V_x + \hat{V}_m^\top V_m) \leq 0.
\end{aligned}
$$

$\square$

**Lemma B.5.** *Under the condition of Theorem 3.1, let*

$$V_x(x,m) = \nabla \mathcal{K}(\tilde{m}) - \lambda x$$

$$V_m(x,m) = -\alpha \nabla f(x) - \gamma m = \frac{\tilde{m} - m}{\varepsilon}$$

*and related*

$$\hat{V}_x(x,m) = \tilde{m} - \nabla \mathcal{K}^*(\lambda x) = -\varepsilon \alpha \nabla f(x) + (1 - \varepsilon \gamma) m - \nabla \mathcal{K}^*(\lambda x),$$

$$\hat{V}_m(x,m) = \nabla \mathcal{K}(\tilde{m}) - \nabla \mathcal{K}(m).$$

*Then we have $\hat{V}_x^\top V_x \geq 0$ and $\hat{V}_m^\top V_m \geq 0$ by Lemma 2.1. Moreover,*

$$\nabla_x H(x,m) = -\eta' \hat{V}_x - \eta V_m$$

$$\nabla_m H(x,m) = -\eta \hat{V}_m + \eta V_x,$$

*where $\eta = \frac{1-\varepsilon\gamma}{1+\varepsilon\lambda}$ and $\eta' = \frac{\gamma+\lambda}{1+\varepsilon\lambda}$. This yields*

$$\frac{\mathrm{d}}{\mathrm{d}t} H(x_t, m_t) = \nabla_x H^\top V_x + \nabla_m H^\top V_m = -(\eta' \hat{V}_x^\top V_x + \eta \hat{V}_m^\top V_m) \leq 0.$$

*Proof.* Let $\eta = \frac{1-\varepsilon\gamma}{1+\varepsilon\lambda}$. We have We have

$$
\begin{aligned}
\nabla_m H(x,m) &= \eta(\nabla \mathcal{K}(m) - \lambda x) \\
&= \eta(\nabla \mathcal{K}(\tilde{m}) - \lambda x + \nabla \mathcal{K}(m) - \nabla \mathcal{K}(\tilde{m})) \\
&= \eta(V_x - \hat{V}_m).
\end{aligned}
$$

$\nabla_x H(x,m)$
$= \alpha \nabla f(x) + \gamma \nabla \mathcal{K}^*(\lambda x) + \eta(\lambda \nabla \mathcal{K}^*(\lambda x) - \lambda m)$
$= \alpha \nabla f(x) + (\gamma + \eta \lambda) \nabla \mathcal{K}^*(\lambda x) - \eta \lambda m$
$= (\gamma + \eta \lambda)(\varepsilon \alpha \nabla f(x) - (1 - \varepsilon \gamma) m + \nabla \mathcal{K}^*(\lambda x)) + (\alpha - (\gamma + \eta \lambda)\varepsilon \alpha) \nabla f(x) - (\eta \lambda - (\gamma + \eta \lambda)(1 - \varepsilon \gamma)) m$
$= \dfrac{\gamma + \lambda}{1 + \varepsilon \lambda}(\varepsilon \alpha \nabla f(x) - (1 - \varepsilon \gamma) m + \nabla \mathcal{K}^*(\lambda x)) + \eta \alpha \nabla f(x) + \eta \gamma m$
$= -\dfrac{\gamma + \lambda}{1 + \varepsilon \lambda} \hat{V}_x - \eta V_m,$

where we used the following identities on $\eta$:

$$(\gamma + \eta \lambda) = \gamma + \frac{1 - \varepsilon \gamma}{1 + \varepsilon \lambda} \lambda = \frac{\gamma + \lambda}{1 + \varepsilon \lambda}$$

$$1 - (\gamma + \eta \lambda)\varepsilon = 1 - \frac{\gamma + \lambda}{1 + \varepsilon \lambda} \varepsilon = \frac{1 - \varepsilon \gamma}{1 + \varepsilon \lambda} = \eta$$

$$\eta \lambda - (\gamma + \eta \lambda)(1 - \varepsilon \gamma) = -\gamma + \frac{\gamma + \lambda}{1 + \varepsilon \lambda} \varepsilon \gamma = \frac{\varepsilon \gamma^2 - \gamma}{1 + \varepsilon \lambda} = -\gamma \eta.$$

$\square$

## B.9 CONSTRAINT ENFORCING: CONTINUOUS TIME

When $\mathcal{K}^*$ can possible take infinite values, the minimization of $H(x, m)$ becomes a constrained optimization. Let $\mathrm{dom}\mathcal{K}^* = \{x \colon \mathcal{K}^*(x) < +\infty\}$. The optimization can be framed as

$$\min_{x,m} H(x, m) \quad s.t. \quad \lambda x \in \mathrm{dom}\mathcal{K}^*.$$

The Lion-$\mathcal{K}$ algorithm would first steer $x_t$ to the region where $\mathcal{K}^*$ has finite values, and then decrease the finite parts of the objective function. In the following, we show that Lion-$\mathcal{K}$ enforces the constraint with a fast linear rate: the distance from $\lambda x_t$ and $\mathrm{dom}\mathcal{K}^*$ decays exponentially fast with time $t$, and once $\lambda x_{t_0} \in \mathrm{dom}\mathcal{K}^*$, then $\lambda x_t$ stays within $\mathrm{dom}\mathcal{K}^*$ for all $t > t_0$.

**Theorem B.6.** *Under the condition of Theorem 3.1, we have*

$$\mathrm{dist}(\lambda x_t, \mathrm{dom}\mathcal{K}^*) \le \exp(\lambda(s - t)) \, \mathrm{dist}(\lambda x_s, \mathrm{dom}\mathcal{K}^*).$$

*Proof.* Define $w_{s \to t} = \exp(\lambda(s - t))$. Integrating $\dot{x}_t = \nabla\mathcal{K}(\tilde{m}_t) - \lambda x_t$, we have

$$\lambda x_t = (1 - w_{s \to t})z_{s \to t} + w_{s \to t}(\lambda x_s), \quad \text{where} \quad z_{s \to t} = \frac{\int_s^t w_{\tau \to t}\nabla\mathcal{K}(\tilde{m}_\tau)\mathrm{d}s}{\int_s^t w_{\tau \to t}d\tau}, \quad \forall 0 \le s \le t.$$

We have $\nabla\mathcal{K}(\tilde{m}_\tau) \in \mathrm{dom}\mathcal{K}^*$ from Lemma B.7 and $\mathrm{dom}\mathcal{K}^*$ is convex. Hence $z_{s \to t}$, as the convex combination of $\{\nabla\mathcal{K}(\tilde{m}_\tau\}_\tau$, belongs to $\mathrm{dom}\mathcal{K}^*$. For any $\epsilon > 0$, let $\lambda\hat{x}_s \in \mathrm{dom}\mathcal{K}^*$ to the point satisfying $\|\lambda\hat{x}_s - \lambda x_s\| \le \mathrm{dist}(\lambda x_s, \mathrm{dom}\mathcal{K}^*) + \epsilon$. Hence,

$$\begin{aligned}
\mathrm{dist}(\lambda x_t, \, \mathrm{dom}\mathcal{K}^*) &= \inf_{z \in \mathrm{dom}\mathcal{K}^*} \|\lambda x_t - z\| \\
&\le \|\lambda x_t - (1 - w_{s \to t})z_{s \to t} - w_{s \to t}\lambda\hat{x}_s)\| \\
&= w_{s \to t} \|\lambda x_s - \lambda\hat{x}_s\| \\
&\le \exp(\lambda(s - t))(\mathrm{dist}(\lambda x_s, \mathrm{dom}\mathcal{K}^*) + \epsilon).
\end{aligned}$$

Taking $\epsilon \to 0$ yields

$$\mathrm{dist}(\lambda x_t, \mathrm{dom}\mathcal{K}^*) \le \exp(\lambda(s - t)) \, \mathrm{dist}(\lambda x_s, \mathrm{dom}\mathcal{K}^*).$$

$\square$

**Lemma B.7.** *Assume $\mathcal{K}$ is proper, closed and convex, and $\mathcal{K}^*$ is the conjugate of $\mathcal{K}$. We have*

$$\partial\mathcal{K}(z) \subseteq \mathrm{dom}\mathcal{K}^*, \quad \forall z \in \mathrm{dom}\mathcal{K}.$$

*Proof.* If $x \in \partial\mathcal{K}(z)$, then $z$ attains the minimum of $\mathcal{K}^*(x) = \sup_z\{x^\top z - \mathcal{K}(z)\}$, suggesting that $\mathcal{K}^*(x) = x^\top z - \mathcal{K}(z) < +\infty$, and hence $x \in \mathrm{dom}\mathcal{K}^*$. $\square$

## B.10 DISCRETE TIME ANALYSIS

**Theorem B.8.** *Assume $f \colon \mathbb{R}^d \to \mathbb{R}$ is L-smooth, and $\mathcal{K} \colon \mathbb{R}^d \to \mathbb{R}$ is closed and convex. Consider the following scheme:*

$$\begin{aligned}
m_{t+1} &= \beta_2 m_t - (1 - \beta_2)\nabla f(x_t) \\
\tilde{m}_{t+1} &= \beta_1 m_t - (1 - \beta_1)\nabla f(x_t) \\
x_{t+1} &= x_t + \epsilon(\nabla\mathcal{K}(\tilde{m}_{t+1}) - \lambda x_{t+1}),
\end{aligned} \qquad (37)$$

*where $\nabla\mathcal{K}$ is a subgradient of $\mathcal{K}$, and $\beta_1, \beta_2 \in (0, 1)$, and $\beta_2 > \beta_1$, and $\epsilon, \lambda > 0$. Let $\mathcal{K}^*$ be the conjugate function of $\mathcal{K}$. Define the following Lyapunov function:*

$$H(x, m) = f(x) + \frac{1}{\lambda}\mathcal{K}^*(\lambda x) + \frac{\beta_1}{\epsilon\lambda(1 - \beta_1) + (1 - \beta_2)}(\mathcal{K}^*(\lambda x) + \mathcal{K}(m) - \lambda x^\top m),$$

*and*

$$\begin{aligned}
\Delta_t^1 &= (\nabla\mathcal{K}(\tilde{m}_{t+1}) - \lambda x_{t+1})^\top(\tilde{m}_{t+1} - \nabla\mathcal{K}^*(\lambda x_{t+1})), \\
\Delta_t^2 &= (\nabla\mathcal{K}(\tilde{m}_{t+1}) - \nabla\mathcal{K}(m_{t+1}))^\top(\tilde{m}_{t+1} - m_{t+1}),
\end{aligned}$$

*where $\nabla\mathcal{K}^*$ is a subgradient of $\mathcal{K}^*$. Then we have $\Delta_t^1 \geq 0$ and $\Delta_t^2 \geq 0$ from Lemma B.9, and*

$$H(x_{t+1}, m_{t+1}) - H(x_t, m_t) \leq -\epsilon(a\Delta_t^1 + b\Delta_t^2) + \frac{L\epsilon^2}{2}\|\nabla\mathcal{K}(\tilde{m}_{t+1}) - \lambda x_{t+1}\|_2^2,$$

*where*

$$a = \frac{\epsilon\lambda\beta_1}{\epsilon\lambda(1-\beta_1)+(1-\beta_2)} + 1 \geq 0, \qquad b = \frac{\beta_1(1-\beta_2)}{(\beta_2-\beta_1)(\epsilon\lambda(1-\beta_1)+(1-\beta_2))} \geq 0.$$

*Hence, a telescoping sum yields*

$$\frac{1}{T}\sum_{t=0}^{T-1} a\Delta_t^1 + b\Delta_t^2 \leq \frac{H(x_0,m_0) - H(x_T,m_T)}{\epsilon T} + \frac{L\epsilon}{2}B_t,$$

*where $B_t = \frac{1}{T}\sum_{t=1}^{T}\|\nabla\mathcal{K}(\tilde{m}_{t+1}) - \lambda x_{t+1}\|_2^2.$*

Note that we used an implicit scheme in the update of $x_t$ in (43). It is equivalent the explicit scheme with an adjusted learning rate:

$$x_{t+1} = x_t + \frac{\epsilon}{1+\epsilon\lambda}(\nabla\mathcal{K}(\tilde{m}_{t+1}) - \lambda x_t).$$

*Proof.* We follow the proof in the continuous-time case to find out a Lyapunov function for the discrete time update in (43). We start with constructing the basic inequalities and work out the Lyapunov function backwardly. From Lemma 2.1, we have

$$(\nabla\mathcal{K}(\tilde{m}_{t+1}) - \lambda x_{t+1})^\top (\nabla\mathcal{K}^*(\lambda x_{t+1}) - \tilde{m}_{t+1}) \leq 0. \tag{38}$$

$$(\nabla\mathcal{K}(\tilde{m}_{t+1}) - \nabla\mathcal{K}(m_{t+1}))^\top (m_{t+1} - \tilde{m}_{t+1}) \leq 0. \tag{39}$$

Taking $a \times Eq.(38) + b \times Eq(39)$ for $a, b \geq 0$, we have

$$(\nabla\mathcal{K}(\tilde{m}_{t+1}) - \lambda x_{t+1})^\top (a(\nabla\mathcal{K}^*(\lambda x_{t+1}) - \tilde{m}_{t+1}) + b(m_{t+1} - \tilde{m}_{t+1})) + \cdots$$
$$+ b(\nabla\mathcal{K}(m_{t+1}) - \lambda x_{t+1})^\top (-m_{t+1} + \tilde{m}_{t+1}) \leq 0.$$

Plugging (43) yields

$$(\nabla\mathcal{K}(\tilde{m}_{t+1}) - \lambda x_{t+1})^\top (a\nabla\mathcal{K}^*(\lambda x_{t+1}) - ((a+b)\beta_1 - b\beta_2)m_t + (a - (a+b)\beta_1 + b\beta_2)\nabla f(x_t))$$
$$- b(\beta_2 - \beta_1)(\nabla\mathcal{K}(m_{t+1}) - \lambda x_{t+1})^\top (m_t + \nabla f(x_t)) \leq 0$$

Define

$$H(x,m) = (a-c)f(x) + \frac{a}{\lambda}\mathcal{K}^*(\lambda x) + \frac{c}{\lambda}\mathcal{K}(m) - cx^\top m, \quad \text{with} \quad c = (a+b)\beta_1 - b\beta_2,$$

and

$$\hat{\nabla}_x H_t = (a-c)\nabla f(x_t) + a\nabla\mathcal{K}^*(\lambda x_{t+1}) - cm_t, \qquad \hat{\nabla}_m H_t = \frac{c}{\lambda}\nabla\mathcal{K}(m_{t+1}) - cx_{t+1}.$$

Then the inequality can be written into

$$\hat{\nabla}_x H_t^\top (\nabla\mathcal{K}(\tilde{m}_{t+1}) - \lambda x_{t+1}) + \hat{\nabla}_m H_t^\top \left(\frac{b(\beta_2 - \beta_1)\lambda}{c}(-m_t - \nabla f(x_t))\right) \leq 0.$$

Plugging the update rule of $x_{t+1} = x_t + \epsilon(\nabla\mathcal{K}(\tilde{m}_{t+1}) - \lambda x_{t+1})$ and $m_{t+1} - m_t = -(1-\beta_2)(m_t + \nabla f(x_t))$, we get

$$\hat{\nabla}_x H_t^\top \left(\frac{x_{t+1} - x_t}{\epsilon}\right) + \hat{\nabla}_m H_t^\top \left(\frac{b(\beta_2 - \beta_1)\lambda}{c(1-\beta_2)}(m_{t+1} - m_t)\right) \leq 0.$$

To make this coincide with the linear approximation of the difference $H(x_{t+1}, m_{t+1}) - H(x_t, m_t)$ (see Lemma B.9), we want

$$\frac{b(\beta_2 - \beta_1)\lambda}{c(1-\beta_2)} = \frac{1}{\epsilon}.$$

On the other hand, to make the coefficient of $f(x)$ in $H(x, m)$ equal to one, we want $a - c = 1$. This yields the following equations on $a, b, c$:

$$c = (a + b)\beta_1 - b\beta_2, \qquad \frac{b(\beta_2 - \beta_1)\lambda}{c(1 - \beta_2)} = \frac{1}{\epsilon}, \qquad a - c = 1, \qquad a, b \geq 0.$$

To solve this, let $c = z\epsilon(\beta_2 - \beta_1)\lambda$ and $b = z(1 - \beta_2)$ for some $z \geq 0$ and plug them together with $a = c + 1$ into the first equations:

$$z\epsilon(\beta_2 - \beta_1)\lambda = (z\epsilon(\beta_2 - \beta_1)\lambda + 1 + z(1 - \beta_2))\beta_1 - z(1 - \beta_2)\beta_2.$$

We get

$$z = \frac{\beta_1}{\epsilon(\beta_2 - \beta_1)\lambda - \epsilon(\beta_2 - \beta_1)\lambda\beta_1 - (1 - \beta_2)\beta_1 + (1 - \beta_2)\beta_2}$$

$$= \frac{\beta_1}{\epsilon\lambda(\beta_2 - \beta_1)(1 - \beta_1) + (1 - \beta_2)(\beta_2 - \beta_1)}$$

$$= \frac{\beta_1}{(\beta_2 - \beta_1)(\epsilon\lambda(1 - \beta_1) + (1 - \beta_2))} \geq 0.$$

Hence

$$b = \frac{\beta_1(1 - \beta_2)}{(\beta_2 - \beta_1)(\epsilon\lambda(1 - \beta_1) + (1 - \beta_2))} \geq 0, \quad c = \frac{\epsilon\lambda\beta_1}{\epsilon\lambda(1 - \beta_1) + (1 - \beta_2)} \geq 0, \quad a = c + 1 \geq 0.$$

In this case, we have

$$H(x, m) = f(x) + \frac{1}{\lambda}\mathcal{K}^*(\lambda x) + c(\mathcal{K}^*(\lambda x) + \mathcal{K}(m) - \lambda x^\top m)$$

$$= f(x) + \frac{1}{\lambda}\mathcal{K}^*(\lambda x) + \frac{\epsilon\lambda\beta_1}{\epsilon\lambda(1 - \beta_1) + (1 - \beta_2)}(\mathcal{K}^*(\lambda x) + \mathcal{K}(m) - \lambda x^\top m),$$

and

$$\hat{\nabla}_x H_t^\top \left(\frac{x_{t+1} - x_t}{\epsilon}\right) + \hat{\nabla}_m H_t^\top \left(\frac{m_{t+1} - m_t}{\epsilon}\right) = -a\Delta_t^1 - b\Delta_t^2 \leq 0.$$

From Lemma B.9, we get

$$H(x_{t+1}, m_{t+1}) - H(x_t, m_t) \leq -\epsilon(a\Delta_t^1 + b\Delta_t^2) + \frac{L}{2}\|x_{t+1} - x_t\|_2^2.$$

$\square$

**Lemma B.9.** *Let* $H(x, m) = f(x) + \mathcal{K}_1(x) + \mathcal{K}_2(m) - \lambda xm$, *where* $f$ *is* $L$-*smooth, and* $\mathcal{K}_1, \mathcal{K}_2$ *are convex functions with subgradient* $\nabla\mathcal{K}_1$ *and* $\nabla\mathcal{K}_2$. *Then*

$$H(x_{t+1}, m_{t+1}) - H(x_t, m_t) \leq \hat{\nabla}_x H_t^\top (x_{t+1} - x_t) + \hat{\nabla}_m H_t^\top (m_{t+1} - m_t) + \frac{L}{2}\|x_{t+1} - x_t\|_2^2,$$

*where*

$$\hat{\nabla}_x H_t = \nabla f(x_t) + \mathcal{K}_1(x_{t+1}) - \lambda m_t$$
$$\hat{\nabla}_m H_t = \mathcal{K}_2(m_{t+1}) - \lambda x_{t+1}.$$

*Note the use of* $x_t$ *vs.* $x_{t+1}$ *and* $m_t$ *vs.* $m_{t+1}$ *in* $\hat{\nabla}_x H_t$ *and* $\hat{\nabla}_m H_t$.

*Proof.* We have

$$f(x_{t+1}) - f(x_t) \leq \nabla f(x_t)^\top (x_{t+1} - x_t) + \frac{L}{2}\|x_{t+1} - x_t\|_2^2$$
$$\mathcal{K}_1(x_{t+1}) - \mathcal{K}_1(x_t) \leq \nabla\mathcal{K}_1(x_{t+1})^\top (x_{t+1} - x_t)$$
$$\mathcal{K}_2(m_{t+1}) - \mathcal{K}_2(m_t) \leq \nabla\mathcal{K}_2(m_{t+1})^\top (m_{t+1} - m_t)$$
$$x_{t+1}^\top m_{t+1} - x_t^\top m_t = m_t^\top (x_{t+1} - x_t) + x_{t+1}^\top (m_{t+1} - m_t).$$

Summing them together yields the result. $\square$

**Theorem B.10.** *Under the same conditions of Theorem 4.1, for any two integers $s \leq t$,*

$$\text{dist}(\lambda x_t, \text{dom}\mathcal{K}^*) \leq \left(\frac{1}{1+\epsilon\lambda}\right)^{t-s} \text{dist}(\lambda x_s, \text{dom}\mathcal{K}^*), \quad \forall s \leq t.$$

*Proof.* Rewriting the update into the explicit form:

$$x_{t+1} = \frac{1}{1+\epsilon\lambda}x_t + \frac{\epsilon}{1+\epsilon\lambda}\nabla\mathcal{K}(\tilde{m}_{t+1}).$$

Unrolling this update yields, with $w_{s\to t} = \left(\frac{1}{1+\epsilon\lambda}\right)^{t-s}$,

$$\lambda x_t = (1 - w_{s\to t})z_{s\to t} + w_{s\to t}x_s, \qquad z_{s\to t} = \frac{\sum_{k=s+1}^{t} w_{k\to t}\nabla\mathcal{K}(\tilde{m}_k)}{\sum_{k=s+1}^{t} w_{k\to t}}.$$

We have $\nabla\mathcal{K}(\tilde{m}_k) \in \text{dom}\mathcal{K}^*$ from Lemma B.7 and $\text{dom}\mathcal{K}^*$ is convex. Hence $z_{s\to t}$, as the convex combination of $\{\nabla\mathcal{K}(\tilde{m}_k)\}_k$, belongs to $\text{dom}\mathcal{K}^*$. For any $\eta > 0$, let $\lambda\hat{x}_s \in \text{dom}\mathcal{K}^*$ to the point satisfying $\|\lambda\hat{x}_s - \lambda x_s\| \leq \text{dist}(\lambda x_s, \text{dom}\mathcal{K}^*) + \eta$. Hence,

$$\begin{aligned}
\text{dist}(\lambda x_t, \text{dom}\mathcal{K}^*) &= \inf_{z\in\text{dom}\mathcal{K}^*} \|\lambda x_t - z\| \\
&\leq \|\lambda x_t - (1 - w_{s\to t})z_{s\to t} + w_{s\to t}\lambda\hat{x}_s)\| \\
&= w_{s\to t}\|\lambda x_s - \lambda\hat{x}_s\| \\
&\leq \left(\frac{1}{1+\epsilon\lambda}\right)^{s-t} (\text{dist}(\lambda x_s, \text{dom}\mathcal{K}^*) + \eta).
\end{aligned}$$

Taking $\eta \to 0$ yields the result.

$\square$

## B.11 ANALYSIS WITH STOCHASTIC GRADIENT FOR LION-$\mathcal{K}$

In this section, we are going to have the convergence analysis of discrete time Lion-$\mathcal{K}$. The proof idea is adapted for section B.10, by defining the same Hamiltonian function, we obtain the bound for $\Delta_t^1$ and $\Delta_t^2$.

Compared with the deterministic case, the main challenge is to bound an additional correlation term due to the stochastic gradient at each iteration $t$:

$$V_t := \text{cov}(g_t, \nabla\mathcal{K}(\tilde{m}_{t+1})) = \text{cov}(g_t, \nabla\mathcal{K}(\beta_1 m_t + (1 - \beta_1)g_t)), \tag{40}$$

where $\text{cov}(X, Y) = \mathbb{E}[(X - \mathbb{E}[X])^\top(Y - \mathbb{E}[Y])]$.

**Definition B.11.** *For a random variable $X$ on $\mathbb{R}^d$, its (trace of) variance $\text{var}(X)$, when exists, is defined as*

$$\text{var}(X) = \mathbb{E}[\|X - \mathbb{E}[X]\|_2^2]$$

**Assumption B.12.** *Assume*

$$\text{var}(g_t) \leq \frac{v_{\max}}{n_{batch}},$$

*where $n_{batch}$ represents the batch size.*

**Assumption B.13.** *$\mathcal{D}$ is the data distribution, the stochastic sample $\xi_t \sim \mathcal{D}$ is i.i.d., given a function $f(x; \xi)$, the gradient $\nabla f(x; \xi)$ is taken with respect to variable $x$, and $\mathbb{E}[\nabla f(x, \xi)] = \nabla f(x)$*

**Theorem B.14.** *Under the assumptions delineated in B.13 and B.12, consider a function $f\colon \mathbb{R}^d \to \mathbb{R}$ that is $L$-smooth. Additionally, let $\mathcal{K}\colon \mathbb{R}^d \to \mathbb{R}$ be a closed and convex function, consider the following scheme:*

$$\begin{aligned}
m_{t+1} &= \beta_2 m_t - (1 - \beta_2)g_t \\
\tilde{m}_{t+1} &= \beta_1 m_t - (1 - \beta_1)g_t \\
x_{t+1} &= x_t + \epsilon(\nabla\mathcal{K}(\tilde{m}_{t+1}) - \lambda x_{t+1}),
\end{aligned} \tag{41}$$

where $g_t = \nabla f(x_t; \xi_t)$ as shown in *B.13*, $m_0, g_1, \ldots, g_t, \ldots$ *are random variables with* $\mathbb{E}[g_t] = \nabla f(x_t)$. $\nabla \mathcal{K}$ *is a weak gradient of* $\mathcal{K}$ *with* $\nabla \mathcal{K}(0) = 0$, $\|\nabla \mathcal{K}(x) - \nabla \mathcal{K}(y)\| \leq L_\mathcal{K} \|x - y\|, \forall x, y \in \mathbb{R}^d$, *and* $\beta_1, \beta_2 \in (0, 1)$, *and* $\beta_2 > \beta_1$, *and* $\epsilon, \lambda > 0$.

*Let* $\mathcal{K}^*$ *be the conjugate function of* $\mathcal{K}$. *Define the following Lyapunov function:*

$$H(x, m) = f(x) + \frac{1}{\lambda}\mathcal{K}^*(\lambda x) + \frac{\beta_1}{\epsilon\lambda(1 - \beta_1) + (1 - \beta_2)}(\mathcal{K}^*(\lambda x) + \mathcal{K}(m) - \lambda x^\top m),$$

*and*

$$\Delta_t^1 = (\nabla \mathcal{K}(\tilde{m}_{t+1}) - \lambda x_{t+1})^\top(\tilde{m}_{t+1} - \nabla \mathcal{K}^*(\lambda x_{t+1})),$$
$$\Delta_t^2 = (\nabla \mathcal{K}(\tilde{m}_{t+1}) - \nabla \mathcal{K}(m_{t+1}))^\top(\tilde{m}_{t+1} - m_{t+1}),$$

*where* $\nabla \mathcal{K}^*$ *is a subgradient of* $\mathcal{K}^*$. *Then we have* $\Delta_t^1 \geq 0$ *and* $\Delta_t^2 \geq 0$ *from Lemma B.9, and*

$$\mathbb{E}\left[H(x_{t+1}, m_{t+1}) - H(x_t, m_t)\right] \leq \mathbb{E}\left[-\epsilon(a\Delta_t^1 + b\Delta_t^2) + \frac{L\epsilon^2}{2}\|\nabla\mathcal{K}(\tilde{m}_{t+1}) - \lambda x_{t+1}\|_2^2\right]$$

$$+ \epsilon \frac{L_\mathcal{K}}{1 + \lambda\epsilon}(1 - \beta_1)\frac{v_{max}}{n_{batch}} + \frac{L_\mathcal{K}}{1 + \lambda\epsilon}\sqrt{\frac{(1 - \beta_2)}{(1 + \beta_2)}}\frac{v_{max}}{n_{batch}}$$

*where*

$$a = \frac{\epsilon\lambda\beta_1}{\epsilon\lambda(1 - \beta_1) + (1 - \beta_2)} + 1 \geq 0, \qquad b = \frac{\beta_1(1 - \beta_2)}{(\beta_2 - \beta_1)(\epsilon\lambda(1 - \beta_1) + (1 - \beta_2))} \geq 0.$$

$v_{\max}, n_{batch}$ *are defined in* *B.12*

*Hence, a telescoping sum yields*

$$\frac{1}{T}\sum_{t=0}^{T-1}\mathbb{E}\left[a\Delta_t^1 + b\Delta_t^2\right] \leq \mathbb{E}\left[\frac{H(x_0, m_0) - H(x_T, m_T)}{\epsilon T} + \frac{L\epsilon}{2}B_t + \frac{C_t}{n_{batch}}\right],$$

*where* $B_t = \frac{1}{T}\sum_{t=1}^T\|\nabla\mathcal{K}(\tilde{m}_{t+1}) - \lambda x_{t+1}\|_2^2$, *and* $C_t = \left(\frac{L_\mathcal{K}}{1 + \lambda\epsilon}(1 - \beta_1) + \frac{L_\mathcal{K}}{1 + \lambda\epsilon}\sqrt{\frac{(1 - \beta_2)}{(1 + \beta_2)}}\right)v_{max}$.

*Proof.* The proof is a simple extended variant of *4.1*. Following the proof of Theorem *B.8*, define

$$H(x, m) = (a - c)f(x) + \frac{a}{\lambda}\mathcal{K}^*(\lambda x) + \frac{c}{\lambda}\mathcal{K}(m) - cx^\top m, \quad \text{with} \quad c = (a + b)\beta_1 - b\beta_2,$$

where

$$a = \frac{\epsilon\lambda\beta_1}{\epsilon\lambda(1 - \beta_1) + (1 - \beta_2)} + 1 \geq 0, \quad b = \frac{\beta_1(1 - \beta_2)}{(\beta_2 - \beta_1)(\epsilon\lambda(1 - \beta_1) + (1 - \beta_2))} \geq 0, \quad c = a - 1.$$

By the definition of $\Delta_t^1, \Delta_t^2$, we have

$a\Delta_t^1 + b\Delta_t^2$

$= a(\nabla\mathcal{K}(\tilde{m}_{t+1}) - \lambda x_{t+1})^\top(\tilde{m}_{t+1} - \nabla\mathcal{K}^*(\lambda x_{t+1}))$

$\qquad + b(\nabla\mathcal{K}(\tilde{m}_{t+1}) - \nabla\mathcal{K}(m_{t+1}))^\top(\tilde{m}_{t+1} - m_{t+1})$

$= (\nabla\mathcal{K}(\tilde{m}_{t+1}) - \lambda x_{t+1})^\top(a(\nabla\mathcal{K}^*(\lambda x_{t+1}) - \tilde{m}_{t+1}) + b(\tilde{m}_{t+1} - m_{t+1}))$

$\qquad + b(\nabla\mathcal{K}(m_{t+1}) - \lambda x_{t+1})^\top(m_{t+1} - \tilde{m}_{t+1})$

$= -(\nabla\mathcal{K}(\tilde{m}_{t+1}) - \lambda x_{t+1})^\top(a\nabla\mathcal{K}^*(\lambda x_{t+1}) - ((a + b)\beta_1 - b\beta_2)m_t + (a - (a + b)\beta_1 + b\beta_2)\nabla f(x_t))$

$\qquad - b\frac{\beta_2 - \beta_1}{1 - \beta_2}\frac{\lambda}{c}(\frac{c}{\lambda}\nabla\mathcal{K}(m_{t+1}) - cx_{t+1})^\top(m_{t+1} - m_t)$

$= -[(a - c)g_t + a\nabla\mathcal{K}^*(\lambda x_{t+1}) - cm_t]^\top(\nabla\mathcal{K}(\tilde{m}_{t+1}) - \lambda x_{t+1})$

$\qquad - \frac{1}{\epsilon}\left[\frac{c}{\lambda}\nabla\mathcal{K}(m_{t+1}) - cx_{t+1}\right]^\top(m_{t+1} - m_t)$

$= -\frac{1}{\epsilon}[(a - c)g_t + a\nabla\mathcal{K}^*(\lambda x_{t+1}) - cm_t]^\top(x_{t+1} - x_t)$

$\qquad - \frac{1}{\epsilon}\left[\frac{c}{\lambda}\nabla\mathcal{K}(m_{t+1}) - cx_{t+1}\right]^\top(m_{t+1} - m_t) \qquad\qquad (42)$

By Lemma B.9,

$$H(x_{t+1}, m_{t+1}) - H(x_t, m_t) \le \hat{\nabla}_x H_t^\top (x_{t+1} - x_t) + \hat{\nabla}_m H_t^\top (m_{t+1} - m_t) + \frac{L}{2} \|x_{t+1} - x_t\|_2^2,$$

where

$$\hat{\nabla}_x H_t = (a - c)\nabla f(x_t) + a\nabla \mathcal{K}^*(\lambda x_{t+1}) - cm_t,$$

$$\hat{\nabla}_m H_t = \frac{c}{\lambda}\nabla \mathcal{K}(m_{t+1}) - cx_{t+1} = \frac{c}{\epsilon\lambda}(\hat{V}_{x,t} - \nabla \mathcal{K}(\tilde{m}_{t+1}) + \nabla \mathcal{K}(m_{t+1}))$$

with

$$V_{x,t} = x_{t+1} - x_t = \epsilon(\nabla \mathcal{K}(\tilde{m}_{t+1}) - \lambda x_{t+1})$$
$$V_{m,t} = m_{t+1} - m_t = -(1 - \beta_2)(g_t - m_t)$$
$$\tilde{m}_{t+1} - m_{t+1} = -(\beta_2 - \beta_1)(g_t - m_t) = -(\beta_2 - \beta_1)V_{m,t}$$
$$\hat{V}_{m,t} = -\nabla \mathcal{K}(\tilde{m}_{t+1}) + \nabla \mathcal{K}(m_{t+1})$$

This gives

$$H(x_{t+1}, m_{t+1}) - H(x_t, m_t)$$
$$\le \hat{\nabla}_x H_t^\top (x_{t+1} - x_t) + \hat{\nabla}_m H_t^\top (m_{t+1} - m_t) + \frac{L}{2} \|x_{t+1} - x_t\|_2^2$$

Hence,

$$H(x_{t+1}, m_{t+1}) - H(x_t, m_t) \le [(a - c)\nabla f(x_t) + a\nabla \mathcal{K}^*(\lambda x_{t+1}) - cm_t]^\top (x_{t+1} - x_t)$$
$$+ \left[\frac{c}{\lambda}\nabla \mathcal{K}(m_{t+1}) - cx_{t+1}\right]^\top (m_{t+1} - m_t) + \frac{L}{2}\|x_{t+1} - x_t\|_2^2$$
$$= [(a - c)g_t + a\nabla \mathcal{K}^*(\lambda x_{t+1}) - cm_t]^\top (x_{t+1} - x_t)$$
$$+ \left[\frac{c}{\lambda}\nabla \mathcal{K}(m_{t+1}) - cx_{t+1}\right]^\top (m_{t+1} - m_t) + \frac{L}{2}\|x_{t+1} - x_t\|_2^2$$
$$+ \epsilon(a - c)(\nabla f(x_t) - g_t)^\top (\nabla \mathcal{K}(\tilde{m}_{t+1}) - \lambda x_{t+1})$$
$$= -\epsilon(a\Delta_t^1 + b\Delta_t^2) + \frac{L}{2}\|x_{t+1} - x_t\|_2^2 \qquad \text{//by equation 44}$$
$$+ \epsilon(a - c)(\nabla f(x_t) - g_t)^\top (\nabla \mathcal{K}(\tilde{m}_{t+1}) - \lambda x_{t+1})$$

It suffices to bound $\mathbb{E}\left[(\nabla f(x_t) - g_t)^\top (\nabla \mathcal{K}(\tilde{m}_{t+1}) - \lambda x_{t+1})\right]$.

Note that

$$\mathbb{E}\left[(\nabla f(x_t) - g_t)^\top (\nabla \mathcal{K}(\tilde{m}_{t+1}) - \lambda x_{t+1})\right]$$
$$= \mathbb{E}\left[(\nabla f(x_t) - g_t)^\top (\frac{1}{1 + \lambda\epsilon}\nabla \mathcal{K}(\tilde{m}_{t+1}) - \frac{\lambda}{1 + \lambda\epsilon}x_t)\right]$$
$$= \frac{1}{1 + \lambda\epsilon}\mathbb{E}\left[(\nabla f(x_t) - g_t)^\top \nabla \mathcal{K}(\tilde{m}_{t+1})\right] + \frac{\lambda}{1 + \lambda\epsilon}\mathbb{E}\left[(\nabla f(x_t) - g_t)^\top x_t\right]$$

By Assumption B.13,

$$\mathbb{E}\left[(\nabla f(x_t) - g_t)^\top \lambda x_t)\right] = \lambda\mathbb{E}_{x_t}\left[\mathbb{E}_{\xi_t}\left[(\nabla f(x_t) - \nabla f(x_t, \xi_t))^\top x_t \mid x_t\right]\right]$$
$$= 0 \qquad \text{//by B.13 } \mathbb{E}[\nabla f(x, \xi)] = \nabla f(x)$$

Next, let us bound $\mathbb{E}\left[(\nabla f(x_t) - g_t)^\top \nabla \mathcal{K}(\tilde{m}_{t+1})\right]$.

$$\mathbb{E}\left[(\nabla f(x_t) - g_t)^\top \nabla \mathcal{K}(\tilde{m}_{t+1})\right] = \mathbb{E}\left[(\nabla f(x_t) - g_t)^\top \nabla \mathcal{K}(\beta_1 m_t - (1 - \beta_1)g_t)\right]$$
$$\le L_\mathcal{K}(1 - \beta_1)\text{var}(g_t) + L_\mathcal{K}\sqrt{\text{var}(\beta_1 m_t) \cdot \text{var}(g_t)} \qquad \text{//by B.18}$$
$$\le L_\mathcal{K}(1 - \beta_1)\frac{v_{max}}{n_{batch}} + L_\mathcal{K}\sqrt{\frac{(1 - \beta_2)}{(1 + \beta_2)}\frac{v_{max}}{n_{batch}}} \qquad \text{//by B.18}$$

Hence,

$$
\begin{aligned}
&\mathbb{E}\left[(\nabla f(x_t) - g_t)^\top (\nabla \mathcal{K}(\tilde{m}_{t+1}) - \lambda x_{t+1})\right] \\
&= \frac{1}{1 + \lambda \epsilon} \mathbb{E}\left[(\nabla f(x_t) - g_t)^\top \nabla \mathcal{K}(\tilde{m}_{t+1})\right] + \frac{\lambda}{1 + \lambda \epsilon} \mathbb{E}\left[(\nabla f(x_t) - g_t)^\top x_{t+1}\right] \\
&\le \frac{L_\mathcal{K}}{1 + \lambda \epsilon}(1 - \beta_1)\frac{v_{max}}{n_{batch}} + \frac{L_\mathcal{K}}{1 + \lambda \epsilon}\sqrt{\frac{(1 - \beta_2)}{(1 + \beta_2)}}\frac{v_{max}}{n_{batch}}
\end{aligned}
$$

$\square$

**Lemma B.15.** *Let $X, Y$ be two $\mathbb{R}^d$-valued random variables with $\mathrm{var}(X) < +\infty$ and $\mathrm{var}(Y) < +\infty$, and assume $\mathcal{K}$ yields a weak derivative $\nabla \mathcal{K}$. We have*

$$
\mathbb{E}[(Y - \mathbb{E}Y)^\top \nabla \mathcal{K}(X + \epsilon Y)] \le L_\mathcal{K}\epsilon \mathrm{var}(Y) + L_\mathcal{K}\mathrm{var}(X) \cdot \mathrm{var}(Y)
$$

*Proof.*

$$
\begin{aligned}
&\mathbb{E}[(Y - \mathbb{E}[Y])^\top \nabla \mathcal{K}(X + \epsilon Y)] \\
&= \mathbb{E}[(Y - \mathbb{E}[Y])^\top \left(\nabla \mathcal{K}(X + \epsilon Y) - \mathcal{K}(\mathbb{E}[X] + \epsilon \mathbb{E}[Y])\right)] \\
&= \sqrt{\mathbb{E}\|Y - \mathbb{E}[Y]\|^2}\sqrt{\mathbb{E}\|\nabla \mathcal{K}(X + \epsilon Y) - \mathcal{K}(\mathbb{E}[X] + \epsilon \mathbb{E}[Y])\|^2} \\
&= \sqrt{\mathbb{E}\|Y - \mathbb{E}[Y]\|^2}\sqrt{L_\mathcal{K}\mathbb{E}\|X + \epsilon Y - \mathbb{E}[X] - \epsilon \mathbb{E}[Y]\|^2} \\
&= L_\mathcal{K}\sqrt{\mathbb{E}\|Y - \mathbb{E}[Y]\|^2}\left(\sqrt{\mathbb{E}\|X - \mathbb{E}[X]\|^2} + \sqrt{\epsilon^2 \|Y - \epsilon\mathbb{E}[Y]\|^2}\right) \\
&= L_\mathcal{K}\epsilon\mathbb{E}\|Y - \mathbb{E}[Y]\|^2 + L_\mathcal{K}\sqrt{\mathbb{E}\|Y - \mathbb{E}[Y]\|^2}\sqrt{\mathbb{E}\|X - \mathbb{E}[X]\|^2} \\
&= L_\mathcal{K}\epsilon\mathrm{var}(Y) + L_\mathcal{K}\sqrt{\mathrm{var}(X) \cdot \mathrm{var}(Y)}
\end{aligned}
$$

$\square$

**Lemma B.16** (Cumulative error of stochastic gradient [4])**.** *Following the same setting in theorem B.14, denote $\delta_l = g_l - \nabla f(x_l)$, for any $k < \infty$ and fixed weight $-\infty < \alpha_1, ..., \alpha_k < \infty$, $\sum_{l=1}^k \delta_l$ is a Martingale. In particular,*

$$
\mathbb{E}\left[\left[\sum_{l=1}^k \alpha_l \delta_l\right]^2\right] \le \sum_{l=1}^k \alpha_l^2 \sigma^2.
$$

*Proof.* We simply check the definition of a Martingale. Denote $Y_k := \sum_{l=1}^k \alpha_l \delta_l$. First, we have that

$$
\begin{aligned}
\mathbb{E}[|Y_k|] &= \mathbb{E}\left[\left|\sum_{l=1}^k \alpha_l \delta_l\right|\right] \\
&\le \sum_l |\alpha_l|\mathbb{E}[|\delta_l|] && \text{triangle inequality} \\
&= \sum_l |\alpha_l|\mathbb{E}[\mathbb{E}[|\delta_l||x_l]] && \text{law of total probability} \\
&\le \sum_l |\alpha_l|\mathbb{E}[\sqrt{\mathbb{E}[\delta_l^2|x_l]}] && \text{Jensen's inequality} \\
&\le \sum_l |\alpha_l|\sigma < \infty
\end{aligned}
$$

Second, again using the law of total probability,

$$\mathbb{E}[Y_{k+1}|Y_1,...,Y_k] = \mathbb{E}\left[\sum_{l=1}^{k+1}\alpha_l\delta_l\Bigg|\alpha_1\delta_1,...,\alpha_k\delta_k\right]$$

$$= Y_k + \alpha_{k+1}\mathbb{E}\left[\delta_{k+1}|\alpha_1\delta_1,...,\alpha_k\delta_k\right]$$

$$= Y_k + \alpha_{k+1}\mathbb{E}\left[\mathbb{E}\left[\delta_{k+1}|x_{k+1},\alpha_1\delta_1,...,\alpha_k\delta_k\right]|\alpha_1\delta_1,...,\alpha_k\delta_k\right]$$

$$= Y_k + \alpha_{k+1}\mathbb{E}\left[\mathbb{E}\left[\delta_{k+1}|x_{k+1}\right]|\alpha_1\delta_1,...,\alpha_k\delta_k\right]$$

$$= Y_k$$

This completes the proof that it is indeed a Martingale. We now make use of the properties of Martingale difference sequences to establish a variance bound on the Martingale.

$$\mathbb{E}[[\sum_{l=1}^{k}\alpha_l\delta_l]^2] = \sum_{l=1}^{k}\mathbb{E}[\alpha_l^2\delta_l^2] + 2\sum_{l<j}\mathbb{E}[\alpha_l\alpha_j\delta_l\delta_j]$$

$$= \sum_{l=1}^{k}\alpha_l^2\mathbb{E}[\mathbb{E}[\delta_l^2|\delta_1,...,\delta_{l-1}]] + 2\sum_{l<j}\alpha_l\alpha_j\mathbb{E}\left[\delta_l\mathbb{E}\left[\mathbb{E}[\delta_j|\delta_1,...,\delta_{j-1}]|\delta_l\right]\right]$$

$$= \sum_{l=1}^{k}\alpha_l^2\mathbb{E}[\mathbb{E}[\mathbb{E}[\delta_l^2|x_l,\delta_1,...,\delta_{l-1}]|\delta_1,...,\delta_{l-1}]] + 0$$

$$= \sum_{l=1}^{k}\alpha_l^2\sigma^2.$$

$\square$

The consequence of this lemma is that we are able to treat $\delta_1,...,\delta_k$ as if they are independent, even though they are not—clearly $\delta_l$ is dependent on $\delta_1,...,\delta_{l-1}$ through $x_l$. By Lemma B.16, we can compute the variance of momentum $m_t$,

$$\text{var}(m_t) = (1-\beta_2)^2\mathbb{E}\left\|\sum_{i=1}^{t}\beta_2^{t-i}\delta_i\right\|^2$$

$$= (1-\beta_2)^2\mathbb{E}\sum_{i=1}^{t}\beta_2^{2t-2i}\|\delta_i\|^2$$

$$= \frac{(1-\beta_2)v_{max}}{(1+\beta_2)n_{batch}}$$

## B.12 ANALYSIS WITH STOCHASTIC GRADIENT LION

**Theorem B.17.** *Under the assumptions delineated in B.13 and B.12, consider a function $f\colon \mathbb{R}^d \to \mathbb{R}$ that is L-smooth. Consider the following scheme:*

$$m_{t+1} = \beta_2 m_t - (1-\beta_2)g_t$$
$$\tilde{m}_{t+1} = \beta_1 m_t - (1-\beta_1)g_t \tag{43}$$
$$x_{t+1} = x_t + \epsilon(sign(\tilde{m}_{t+1}) - \lambda x_{t+1}),$$

*where $g_t = \nabla f(x_t;\xi_t)$ as shown in B.13, $m_0, g_1,\ldots,g_t,\ldots$ are random variables with $\mathbb{E}[g_t] = \nabla f(x_t)$. $\beta_1, \beta_2 \in (0,1)$, and $\beta_2 > \beta_1$, and $\epsilon, \lambda > 0$.*

*Define the following Lyapunov function:*

$$H(x,m) = f(x) + \frac{1}{\lambda}\|\lambda x\|^* + \frac{\beta_1}{\epsilon\lambda(1-\beta_1)+(1-\beta_2)}(\|\lambda x\|^* + \|m\| - \lambda x^\top m),$$

*and*

$$\Delta_t^1 = (sign(\tilde{m}_{t+1}) - \lambda x_{t+1})^\top(\tilde{m}_{t+1} - sign^*(\lambda x_{t+1})),$$
$$\Delta_t^2 = (sign(\tilde{m}_{t+1}) - sign(m_{t+1}))^\top(\tilde{m}_{t+1} - m_{t+1}),$$

*where $sign^*$ is a subgradient of $\mathcal{K}^*$. Then we have $\Delta_t^1 \geq 0$ and $\Delta_t^2 \geq 0$ from Lemma B.9, and*

$$\mathbb{E}\left[H(x_{t+1}, m_{t+1}) - H(x_t, m_t)\right] \leq \mathbb{E}\left[-\epsilon(a\Delta_t^1 + b\Delta_t^2) + \frac{L\epsilon^2}{2}\left\|sign(\tilde{m}_{t+1}) - \lambda x_{t+1}\right\|_2^2\right] + \epsilon\frac{1}{1+\lambda\epsilon}\frac{\sqrt{d \cdot v_{max}}}{\sqrt{n_{batch}}}$$

*where*

$$a = \frac{\epsilon\lambda\beta_1}{\epsilon\lambda(1-\beta_1)+(1-\beta_2)} + 1 \geq 0, \qquad b = \frac{\beta_1(1-\beta_2)}{(\beta_2-\beta_1)(\epsilon\lambda(1-\beta_1)+(1-\beta_2))} \geq 0.$$

*$v_{max}$, $n_{batch}$ are defined in B.12*

*Hence, a telescoping sum yields*

$$\frac{1}{T}\sum_{t=0}^{T-1}\mathbb{E}\left[a\Delta_t^1 + b\Delta_t^2\right] \leq \mathbb{E}\left[\frac{H(x_0, m_0) - H(x_T, m_T)}{\epsilon T} + \frac{L\epsilon}{2}B_t + \frac{1}{1+\lambda\epsilon}\frac{\sqrt{d \cdot v_{max}}}{\sqrt{n_{batch}}}\right],$$

*where $B_t = \frac{1}{T}\sum_{t=1}^{T}\left\|sign(\tilde{m}_{t+1}) - \lambda x_{t+1}\right\|_2^2$*

*Proof.* Define

$$H(x, m) = (a-c)f(x) + \frac{a}{\lambda}\|\lambda x\|^* + \frac{c}{\lambda}\|m\| - cx^\top m, \qquad \text{with} \qquad c = (a+b)\beta_1 - b\beta_2,$$

where

$$a = \frac{\epsilon\lambda\beta_1}{\epsilon\lambda(1-\beta_1)+(1-\beta_2)} + 1 \geq 0, \quad b = \frac{\beta_1(1-\beta_2)}{(\beta_2-\beta_1)(\epsilon\lambda(1-\beta_1)+(1-\beta_2))} \geq 0, \quad c = a - 1.$$

By the definition of $\Delta_t^1, \Delta_t^2$, we have

$$a\Delta_t^1 + b\Delta_t^2$$
$$= a(sign(\tilde{m}_{t+1}) - \lambda x_{t+1})^\top(\tilde{m}_{t+1} - sign^*(\lambda x_{t+1}))$$
$$\qquad + b(sign(\tilde{m}_{t+1}) - sign(m_{t+1}))^\top(\tilde{m}_{t+1} - m_{t+1})$$
$$= (sign(\tilde{m}_{t+1}) - \lambda x_{t+1})^\top(a(sign^*(\lambda x_{t+1}) - \tilde{m}_{t+1}) + b(\tilde{m}_{t+1} - m_{t+1}))$$
$$\qquad + b(sign(m_{t+1}) - \lambda x_{t+1})^\top(m_{t+1} - \tilde{m}_{t+1})$$
$$= -(sign(\tilde{m}_{t+1}) - \lambda x_{t+1})^\top(asign^*(\lambda x_{t+1}) - ((a+b)\beta_1 - b\beta_2)m_t + (a - (a+b)\beta_1 + b\beta_2)\nabla f(x_t))$$
$$\qquad - b\frac{\beta_2 - \beta_1}{1-\beta_2}\frac{\lambda}{c}(\frac{c}{\lambda}sign(m_{t+1}) - cx_{t+1})^\top(m_{t+1} - m_t)$$
$$= -\left[(a-c)g_t + asign^*(\lambda x_{t+1}) - cm_t\right]^\top(sign(\tilde{m}_{t+1}) - \lambda x_{t+1})$$
$$\qquad - \frac{1}{\epsilon}\left[\frac{c}{\lambda}sign(m_{t+1}) - cx_{t+1}\right]^\top(m_{t+1} - m_t)$$
$$= -\frac{1}{\epsilon}\left[(a-c)g_t + asign^*(\lambda x_{t+1}) - cm_t\right]^\top(x_{t+1} - x_t)$$
$$\qquad - \frac{1}{\epsilon}\left[\frac{c}{\lambda}sign(m_{t+1}) - cx_{t+1}\right]^\top(m_{t+1} - m_t) \qquad (44)$$

By Lemma B.9,

$$H(x_{t+1}, m_{t+1}) - H(x_t, m_t) \leq \hat{\nabla}_x H_t^\top(x_{t+1} - x_t) + \hat{\nabla}_m H_t^\top(m_{t+1} - m_t) + \frac{L}{2}\|x_{t+1} - x_t\|_2^2,$$

where

$$\hat{\nabla}_x H_t = (a-c)\nabla f(x_t) + asign^*(\lambda x_{t+1}) - cm_t,$$
$$\hat{\nabla}_m H_t = \frac{c}{\lambda}sign(m_{t+1}) - cx_{t+1} = \frac{c}{\epsilon\lambda}(\hat{V}_{x,t} - sign(\tilde{m}_{t+1}) + sign(m_{t+1}))$$

with

$$V_{x,t} = x_{t+1} - x_t = \epsilon(sign(\tilde{m}_{t+1}) - \lambda x_{t+1})$$
$$V_{m,t} = m_{t+1} - m_t = -(1 - \beta_2)(g_t - m_t)$$
$$\tilde{m}_{t+1} - m_{t+1} = -(\beta_2 - \beta_1)(g_t - m_t) = -(\beta_2 - \beta_1)V_{m,t}$$
$$\hat{V}_{m,t} = -sign(\tilde{m}_{t+1}) + sign(m_{t+1})$$

This gives

$$H(x_{t+1}, m_{t+1}) - H(x_t, m_t)$$
$$\leq \hat{\nabla}_x H_t^\top (x_{t+1} - x_t) + \hat{\nabla}_m H_t^\top (m_{t+1} - m_t) + \frac{L}{2} \|x_{t+1} - x_t\|_2^2$$

Hence,

$$H(x_{t+1}, m_{t+1}) - H(x_t, m_t) \leq [(a - c)\nabla f(x_t) + a sign^*(\lambda x_{t+1}) - cm_t]^\top (x_{t+1} - x_t)$$
$$+ \left[\frac{c}{\lambda} sign(m_{t+1}) - cx_{t+1}\right]^\top (m_{t+1} - m_t) + \frac{L}{2} \|x_{t+1} - x_t\|_2^2$$
$$= [(a - c)g_t + a sign^*(\lambda x_{t+1}) - cm_t]^\top (x_{t+1} - x_t)$$
$$+ \left[\frac{c}{\lambda} sign(m_{t+1}) - cx_{t+1}\right]^\top (m_{t+1} - m_t) + \frac{L}{2} \|x_{t+1} - x_t\|_2^2$$
$$+ \epsilon(a - c)(\nabla f(x_t) - g_t)^\top (sign(\tilde{m}_{t+1}) - \lambda x_{t+1})$$
$$= -\epsilon(a\Delta_t^1 + b\Delta_t^2) + \frac{L}{2} \|x_{t+1} - x_t\|_2^2 \quad \text{//by equation 44}$$
$$+ \epsilon(a - c)(\nabla f(x_t) - g_t)^\top (sign(\tilde{m}_{t+1}) - \lambda x_{t+1})$$

It suffices to bound $\mathbb{E}\left[(\nabla f(x_t) - g_t)^\top (sign(\tilde{m}_{t+1}) - \lambda x_{t+1})\right]$.

Note that

$$\mathbb{E}\left[(\nabla f(x_t) - g_t)^\top (sign(\tilde{m}_{t+1}) - \lambda x_{t+1})\right]$$
$$= \mathbb{E}\left[(\nabla f(x_t) - g_t)^\top \left(\frac{1}{1 + \lambda\epsilon} sign(\tilde{m}_{t+1}) - \frac{\lambda}{1 + \lambda\epsilon} x_t\right)\right]$$
$$= \frac{1}{1 + \lambda\epsilon}\mathbb{E}\left[(\nabla f(x_t) - g_t)^\top sign(\tilde{m}_{t+1})\right] + \frac{\lambda}{1 + \lambda\epsilon}\mathbb{E}\left[(\nabla f(x_t) - g_t)^\top x_t\right]$$

By Assumption B.13,

$$\mathbb{E}\left[(\nabla f(x_t) - g_t)^\top \lambda x_t)\right] = \lambda\mathbb{E}_{x_t}\left[\mathbb{E}_{\xi_t}\left[(\nabla f(x_t) - \nabla f(x_t, \xi_t))^\top x_t \mid x_t\right]\right]$$
$$= 0 \quad \text{//by B.13} \quad \mathbb{E}[\nabla f(x, \xi)] = \nabla f(x)$$

Next, we can use B.18 to bound $\mathbb{E}\left[(\nabla f(x_t) - g_t)^\top sign(\tilde{m}_{t+1})\right]$.

$$\mathbb{E}\left[(\nabla f(x_t) - g_t)^\top sign(\tilde{m}_{t+1})\right] = \mathbb{E}\left[(\nabla f(x_t) - g_t)^\top sign(\beta_1 m_t - (1 - \beta_1)g_t)\right]$$
$$\leq \sqrt{d \cdot \mathrm{var}(g_t)} \quad \text{//by B.18}$$
$$\leq \sqrt{\frac{d \cdot v_{max}}{n_{batch}}} \quad \text{//by B.12}$$

Hence,

$$\mathbb{E}\left[(\nabla f(x_t) - g_t)^\top (sign(\tilde{m}_{t+1}) - \lambda x_{t+1})\right]$$
$$= \frac{1}{1 + \lambda\epsilon}\mathbb{E}\left[(\nabla f(x_t) - g_t)^\top sign(\tilde{m}_{t+1})\right] + \frac{\lambda}{1 + \lambda\epsilon}\mathbb{E}\left[(\nabla f(x_t) - g_t)^\top x_{t+1}\right]$$
$$\leq \frac{1}{1 + \lambda\epsilon}\sqrt{\frac{d \cdot v_{max}}{n_{batch}}}$$

$\square$

**Lemma B.18.** *Let $X, Y$ be two $\mathbb{R}^d$-valued random variables with $\mathrm{var}(Y) < +\infty$, and assume $\mathcal{K}$ yields a weak derivative $sign$. We have $\mathbb{E}[(Y - \mathbb{E}Y)^\top sign(X + \epsilon Y)] \leq \sqrt{d\,\mathrm{var}(Y)}$*

*Proof.*

$$\mathbb{E}[(Y - \mathbb{E}[Y])^\top sign(X + \epsilon Y)] \leq \mathbb{E}[|Y - \mathbb{E}[Y]|] \leq \sqrt{d \cdot \mathbb{E}[\|Y - \mathbb{E}[Y]\|^2]} = \sqrt{d \cdot \mathrm{var}(Y)}$$

$\square$

