# OpenReview forum: "Lion Secretly Solves a Constrained Optimization: As Lyapunov Predicts"
_ICLR.cc/2024/Conference — ICLR 2024 spotlight_

### Official Review · Reviewer_GggD · 2023-11-01

**Soundness:** 3 good
**Presentation:** 4 excellent
**Contribution:** 4 excellent
**Rating:** 8
**Confidence:** 3

**Summary:**

This paper analyzes the recently introduced Lion optimizer, which was found previously in the literature through program search, and shows how it fits into a general family of optimizers, Lion-$\mathcal{K}$ which are shown to optimize a specific regularized/constrained version of the original cost function. This optimizer also recovers several known optimization methods from the literature such as mirror descent, momentum or weight decay. The analysis of this optimizer is done by proposing a new general of Lyapunov function yielded by the dynamics of Lion-$\mathcal{K}$.  Results are provided in the continuous time and discrete time setting. Additional experimental results are provided to illustrate the training dynamics.

**Strengths:**

### Originality

I believe the work is original, as it is, up to my knowledge, the first to propose such a general general Lion-$\mathcal{K}$ algorithm along with the corresponding Lyapunov function for its dynamics. In addition the derivation of such a Lyapunov function seems non-trivial, and I believe it will be of interest to the community.

### Quality

I believe the quality is good, as theorems are clearly stated with their assumptions, and full proofs are well detailed in appendix, and with an overview in the main body. Particular care was also taken in the experimental analysis to verify the theoretical results in practice.

### Clarity

The work is clear, theorems are clearly stated as well as their assumptions.

### Significance

I think this work is very significant, since Lion is currently a state of the art optimizer in deep learning, therefore analyzing its dynamics is of very high importance to the machine learning community. Additionally, the discussion on the relationship of Lion-$\mathcal{K}$ with other methods from the literature such as Mirror Descent, Momentum etc makes this analysis even more relevant and general.

**Weaknesses:**

I just have a question regarding the impact of assuming differentiability of $\mathcal{K}$ in the continuous-time analysis (see question below).

I also noticed just a few minor remarks/typos:
- “Going beyond Lion, difference $\mathcal{K}$” —> “Going beyond Lion, different $\mathcal{K}$”
- “One can think Lion” —> “One can think of Lion”
- “Gradient Enhacenment” —> “Gradient Enhancement”
- “Section 3 analysis” —> “Section 3 analyzes”
- Appendix B.6 : “Becomes a contraint optimization” —> “Becomes a constrained optimization”
- Although I think the Figures 1-3 are nice, unless I am mistaken those are not mentioned in the main text: I believe it could be good to mention them, for instance after the discussion on the different phases of training.

**Questions:**

I just have one question regarding the impact of the non-differentiability of $\mathcal{K}$:

(a) In the continuous-time setting, it is assumed that $\mathcal{K}$ is differentiable. Then in such case the convergence to a stationarity point of the constrained/penalized function for the (continuous-time) algorithm is established in Theorem 3.1.

(b) Then, a discrete-time analysis is provided, which allows $\mathcal{K}$ not to be differentiable, but however in such case I think the result is a little bit more difficult to interpret: indeed, the only result provided is  Thm 4.1, which is about $H(x_{t+1}, m_{t+1}) - H(x_{t}, m_t)$ (or, alternatively, about the sum of deltas), but unless I am mistaken, there is no result that looks similar to convergence to a stationary point of the constrained objective (there is also Theorem B.9 in Appendix but that one describes only the first phase of the dynamics). Therefore, I think it is a bit hard to verify from reading such theorem whether the conclusions of the discrete-time case (which correspond to the algorithm used in practice) will actually match the conclusions from the continuous-time case.

More precisely, those two theoretical results above are interesting in themselves, but I am just wondering to what extent they can indeed successfully predict the actual behaviour of the Lion-$\mathcal{K}$ method: indeed, as mentioned in the paper, Lyapunov analyses need $\mathcal{K}$ to be differentiable.  When the function $\mathcal{K}$ is not differentiable (as it is the case for the $\ell_1$ norm which is considered in the paper), analyses can break. But I believe this is more than just a technicality, i.e. I think the behaviour of algorithms for differentiable vs. non-differentiable $\mathcal{K}$ may be quite different: indeed, in the simple case of mirror descent (section 3.1), for instance, writing the discrete Mirror Descent update naively with $\mathcal{K}$ taken as the $\ell_1$ norm gives an algorithm of the form $x_{t+1} \leftarrow \text{sign}(x_t - \nabla f(x_t))$, which clearly does not converge in general to $min_x f(x)$ as iterates remain restricted in $\\{ +1, 0, -1\\}^{d}$ (the same problem would also happen for the continuous case Mirror Descent). But however, I think that having a non-differentiable $\mathcal{K}$ for the usual discrete-time Frank-Wolfe is OK.

Therefore, I believe it would be good to elaborate more (and/or give more references), in the continous case, on why the analysis of Lion-$\mathcal{K}$ would still somehow be valid or almost valid for non-differentiable $\mathcal{K}$, and/or, in the discrete-case setting, to give more details or a more intuitive reading of the result in Theorem 4.1. I think this would allow the reader to further confirm that what Lion-$\mathcal{K}$ is doing is indeed constrained/penalized minimization, without worrying that issues similar to the ones of Mirror Descent happen in the case of Lion-$\mathcal{k}$.

---

> ### Author Response · Authors · 2023-11-23
>
> Dear Reviewer, we thank you for your valuable comments. We have addressed the typos and further refined the draft. Below are answers to your questions on the differentiability of $\mathcal{K}$.
>
> 1) The main difference between the continuous and discrete-time results is that $\Delta(x,m)$, which is a measure of stationarity of the problem, decreases with $O(1/t)$ rate in the continuous time, while $O(1/(\epsilon T) + \epsilon)$ in the discrete time, where $T$ denotes the number of steps. Hence, the continuous-time and discrete-time results are parallel and consistent, in the sense that we recover the continuous-time result when taking the step size $\epsilon$ to zero (with $t = \epsilon T$).
>
> 3) If we assume $\mathcal{K}$ is smooth (with bounded Hessian), it is possible to improve the discrete-time rate to $O(1/(\epsilon T))$. Hence, the impact of the non-differentiability is the $O(\epsilon)$ term, which suggests that the algorithm converges up to an $\epsilon$ accuracy. This is a typical phenomenon in optimization with non-smooth objectives (like sub-gradient descent) or non-smooth update (like signed GD). Because in practice the step size is small or decaying, this may not have a big impact on practical performance.
>
> 4) The requirement for differentiability in our statement of continuous-time results is mainly for the simplicity of the presentation. We can state a continuous-time result for non-smooth $\mathcal{K}$ using non-smooth versions of Lyapunov theory, but we would need to recall sophisticated technical tools such as Filippov's differential inclusion and Clarke's generalized gradient. See e.g., *Daniel Shevitz and Brad Padent, Lyapunov Stability Theory of Nonsmooth Systems, 1993.*
>
> 5) The mirror descent with $\ell_1$ norm reduces to Signed GD: $x_{t+1} = x_t - \epsilon sign(\nabla f(x_t))$, which is different from $x_{t+1} = sign(x_t - \nabla f(x_t))$. Hence it is the update (not the value of $x_t$) that is constrained in $[-1,1]$. With standard argument, one can show that $\min_{s\leq T}\|\nabla f(x_s)\|_1 = O(1/(\epsilon T) + \epsilon)$, which is consistent with our result.

---

> ### Comment · Reviewer_GggD · 2023-11-24
> **Response to Authors**
>
> **[[Sorry for the late reply, I realized that Authors were not included by default when I answered on the 24th, and therefore have updated the visibility now]]**
>
>  Dear Authors, thank you for your answer.
>
> Regarding the Mirror Descent case, I think your response has adressed my concern, I just have a small question to confirm my understanding. First of all, I apologize, I was mistaken in my previous review when writing the Mirror Descent update in the discrete case: indeed, it should actually be something like 8.1.3 in [1]: (1): (a)$x\_{t} = \nabla \mathcal{K}(y\_{t})$ and (b): $y\_{t+1} = y\_t - \eta \nabla f(x\_{t})$ (for some learning rate $\eta$). But there, we see that still, using a non-differentiable function $\mathcal{K}$ in a naive way would result in having iterates in $\\{-1, 0, 1\\}$, if I am not mistaken (if $\mathcal{K}$ is the $\ell\_1$ norm) ?
> However, I have now understood by reading the revision and in the light of Appendix B.4 that actually Lion-$\mathcal{K}$ method does not exactly recover Mirror Descent, but rather a *variant* of Mirror Descent, i.e. it does not recover the update rule (1) (which would be a naive usage of $\mathcal{K}$ as a dual of the distance generating function for Mirror Descent), but it recovers some other algorithm, is that right ?
> If so, then that would indeed discard the pathological case of iterates being in $\\{-1, 0, +1 \\}$.
>
> Let me know if my understanding is correct (if answering on Openreview is still possible given that the deadline has passed), but in any case, I have increased my confidence score in the light of the response you gave, thanks a lot again.
>
>
> [1] Vanli, Nuri Denizcan. Large-Scale Optimization Methods: Theory and Applications. Diss. Massachusetts Institute of Technology, 2021.

---

### Official Review · Reviewer_2PBx · 2023-11-03

**Soundness:** 2 fair
**Presentation:** 2 fair
**Contribution:** 3 good
**Rating:** 6
**Confidence:** 3

**Summary:**

This paper provides a theoretical understanding of the Lion optimizer, a new optimizer discovered by program search. The authors introduced a general class of optimizers, Lion-$\mathcal{K}$, and showed that these optimizers minimize the training loss $f(x)$ with an additional constraint on $\mathcal{K}^*(x)$ via a Lyapunov based analysis. The Lion optimizer corresponds to the case where $\mathcal{K}
$ equals the $L_1$-norm of $x$, thereby imposing a constraint on $\|\|x\|\|_{\infty}$. Extending beyond Lion, the proposed Lion-$\mathcal{K}$ also encompasses algorithms such as Polyak/Nesterov Momentum, Singed Momentum, and (Accelerated) Mirror Descent.

**Strengths:**

The main idea of this paper is novel and interesting. It offers a fresh perspective on the theoretical understanding of Lion and sheds light on the role of the key components, such as the (decoupled) weight decay and the gradient enhancement (the Nesterov trick). The proposed family, Lion-$\mathcal{K}$, also encompasses a wide range of optimizers.

**Weaknesses:**

1. This manuscript appears to be incomplete. Notably, there is a lack of discussion on related works, and the empirical results presented in Figures 1-3 are neither interpreted nor mentioned, even though there seems to be sufficient space for such discussions. I strongly recommend that the authors enhance the completeness and rigor of this paper via:
   - Interpretation of the empirical results,
   - Ablation studies on the key components of Lion to justify the theory,
   - Empirical validation of (at least some) algorithms listed in Table 2.

2. This paper does not connect the superior performance of Lion or other Lion-$\mathcal{K}$ optimizers to their main finding. How does this additional constraint help optimization or generalization?


3. (Minor) Some typos:
    - In the definition of the conjugate function below eq (4), the subscript should be $z$ instead of $x$.
    - Above Lemma 2.1: crucial rule -> crucial role. In Lemma 2.1, $\nabla \mathcal{K},\nabla \mathcal{K} \to \nabla \mathcal{K},\nabla \mathcal{K^*} $.

**Questions:**

See weaknesses.

---

> ### Author Response · Authors · 2023-11-23
> **Rebuttal (part 1/3)**
>
> Dear reviewer, thank you very much for your comments. We have addressed the typos and further refined the draft.
>
> ### 1. Interpretation of empirical results
>
> **Ans:** We have added further interpretations of the empirical results in the revised draft as mentioned in general response. The following is the interpretation the results in Figure 1-3.
>
> - **Fig 1:** The trajectories of Lion with weight decay $\lambda = 1.5$ and $\lambda = 0.6$ and corresponding loss curve and $||x||_\infty$ curve. Observation: the converging points of those 2 trajectories are within the feasible region (specifically on the boundary of the feasible region). We use this figure to verify 2 things:
>     1. Weight decay introduces a constraint effect.
>     2. 2 phases of training dynamics: (first, exponentially decay to feasible region, then minimize objective function)
>
> - **Fig 2:** We use this figure to show phase I is very fast and the constraint actually exists. The histograms of network parameters trained by Lion with $\lambda = 10$, the initialization of network parameters is Kaiming normal, from the leftmost plot, in the beginning, not all params satisfy the constraint (red vertical lines: $x\leq 1 / \lambda$). After 150 iterations of training, the constraint is almost satisfied, and at the 250-th iteration, the constraint is fully satisfied.
>
> - **Fig 3:** This figure is a stack version of Figure 2 but with different initialization methods and weight decay ($\lambda$) levels. Stacked histograms of network parameters in ResNet on the CIFAR-10 dataset across iterations. We stacked the histogram plots from iteration 0 (back) to 190 (front). With weight decay $\lambda$ equals to 0, there is no constraint effect.
>
> ### 2. Empirical validation of algorithms listed in Table 2
>
> **Ans:** As we mentioned in the general response, we have **Experiment II (Toy Example with Different $\mathcal{K}$):**
> - **Section & Figure Reference:** Section 5, Page 10 -11, Figure 1 and Figure 5.
> - **Summary:** We examine Lion's behavior with different $\mathcal{K}$ values in a similar toy example from Figure 1. This demonstrates that the observed behavior aligns well with our theoretical predictions.
>
>
> ### 3. Connect the superior performance of Lion or other Lion-K optimizers to their main finding.
>
> **Ans:** We have added experiments mentioned in the general response:
>
> **Experiment I (The Implication of Weight Decay):**
> - **Section & Figure Reference:** Page 4-5, Figure 1 and Figure 4.
> - **Summary:** We discuss the optimal weight decay $\lambda$ leading to fast convergence without sacrificing training loss. We demonstrate that a larger $\lambda$ generally accelerates learning, but performance drops when $\lambda$ exceeds a critical value $\lambda_0$. This finding underscores the importance of tuning $\lambda$ for Lion.
>
> **Experiment III (Lion-$\ell_p$ on ImageNet and Language Modeling):**
> - **Section & Figure Reference:** Section 5, Figure 7.
> - **Summary:** We evaluate Lion-$\ell_p$ with $p \in \{1.0, 1.2, 1.4, 1.6, 1.8, 2.0\}$ on ImageNet and Language Modeling tasks using ResNet-50, ViT-16B, and GPT-2. The results are intriguing: Lion-$\ell_p$ with smaller $p$ values performs better for ViT and GPT-2, but the opposite trend is observed with ResNet-50 on ImageNet. This suggests a potential architecture-dependence of the optimal $\mathcal{K}$, marking an interesting avenue for future research.
>
> ### 4. How does this additional constraint help optimization or generalization?
>
> **Ans:** There is a trade-off effect. Adding a weight decay term introduces a constraint. Adding the constraint force drives the solution faster towards the region.
>
> ### 5. Why Should Lion Decay Weight?
>
> **Ans:** From the analysis above, the role of weight decay $\lambda$ in Lion is two-fold:
>
> a) It alternates the solution if $\lambda$ is large and the constraint $||x||_\infty$ $\leq$ $1/\lambda$
>
> is strong enough to exclude the unconstrained minimum $x_{unc}^*$ of $f(x)$. This may improve the generalization and stability of the solution while reducing the training loss (Figure 4 and Figure 7).
>
> b) If $\lambda$ is sufficiently small to include the unconstrained minimum $x_{\text{unc}}^*$ in the constrained set, it does not alter the final solution. In this case, the main role of weight decay is to speed up the convergence because the Phase 1 brings the solution into the constrained set with a linear rate. Hence, the ideal choice of $\lambda$ is $\lambda$ $=$ $1/||x_{unc}^*||_\infty$

---

> ### Author Response · Authors · 2023-11-23
> **Rebuttal (part 2/3)**
>
> ### Related work
>
> We have a related work part on page 2, also listed other related algorithms in Table 1.
>
> 1) **Sign Reshaper:**
> The use of the $sign(\cdot)$ function for update, similar to signed gradient descent and signed momentum [1, 2], can be viewed as an extreme way of normalizing the magnitude of the coordinate-wise updates. It is closely related to normalized gradient [3, 4] and adaptive gradient methods such as Adam [5] and RMSprop [6]. Note that Adam can be viewed as signed momentum with an adaptive variance-based step size, which might be the key factor explaining the gap between Adam and SGD [7].
>
> 2) **Gradient Enhancement:**
> When using $\beta_2 > \beta_1$, the importance of the current gradient \(g_t\) is increased compared to the exponential moving average in standard Polyak momentum update. It can be shown that Polyak momentum with this gradient enhancement results in Nesterov momentum, and leads to the well-known acceleration phenomenon [7].
>
> 3) **Decoupled Weight Decay:**
> The weight decay term $\lambda x_t$ outside of the gradient and $sign(\cdot)$. This idea of decoupled weight decay is what makes AdamW [8] significantly outperform vanilla Adam in training large AI models.
>
> **[1]** Bernstein et al., 2018 - signSGD: Compressed Optimisation for Non-Convex Problems
>
> **[2]** Crawshaw et al., 2022 - Robustness to Unbounded Smoothness of Generalized signSGD
>
> **[3]** Levy, 2016 - The Power of Normalization: Faster Evasion of Saddle Points
>
> **[4]** Murray et al., 2019 - Revisiting Normalized Gradient Descent: Fast Evasion of Saddle Points
>
> **[5]** Kingma and Ba, 2014 - Adam: A Method for Stochastic Optimization
>
> **[6]** Tieleman and Hinton, 2012 - RMSprop
>
> **[7]** Shi et al., 2021 - Understanding the Acceleration Phenomenon via High-Resolution Differential Equations
>
> **[8]** Loshchilov and Hutter, 2017 - Decoupled Weight Decay Regularization

---

> ### Author Response · Authors · 2023-11-23
> **Rebuttal (part 3/3)**
>
> **Key components of Lion**
>
> A. The double-$\beta$ scheme
> Up to the rebuttal period, the authors have found that the double-$\beta$ scheme, in fact, is closely related to Nesterov accelerated gradient (NAG), which is known to have the gradient correction effect that accelerates and stabilize learning [1,2]. We believe this observation will promote more innovations in the intersection of Lion and NAG.
>
> For derivation, consider Nesterov Momentum: $$ m_t \leftarrow \beta m_{t-1} + (1 - \beta) \nabla f(x_{t-1} - \alpha m_{t-1}),~~~~x_t \leftarrow x_{t-1} - \alpha m_t $$
>
> Taking $\theta_t = x_t - \alpha m_t$, we have
>
> $$ m_t \leftarrow \beta m_{t-1} + (1 - \beta) \nabla f(\theta_{t-1}),~~~~\theta_t \leftarrow \theta_{t-1} - \alpha \bigg((2\beta - 1) m_{t-1} + 2(1 - \beta)\nabla f(\theta_{t-1})\bigg) $$
>
> From above, one can easily see that Nesterov momentum is just the double-$\beta$ scheme with $\beta_1 = 2\beta - 1$. In practice (as in PyTorch's official implementation), it is optional to have the $(1 - \beta)$ damping term. Without $(1- \beta)$, it becomes $$ \theta_t \leftarrow \theta_{t-1} - \alpha \bigg((2\beta - 1) m_{t-1} + 2\nabla f(\theta_{t-1})\bigg) $$
>
> We should note that in both cases above and other variants of implementations, NAG is only parameterized by a single $\beta$, rather than two decoupled $\beta_1, \beta_2$, and hence miss the sweet spot of $\beta_1=0.9$ and $\beta_2 = 0.99$ found by Lion.
>
> References:
>
> [1] Understanding the Acceleration Phenomenon via High-Resolution Differential Equations.
>
> [2] On the importance of initialization and momentum in deep learning.
>
> B. The sign function
> We think that the main effect of the sign function is to normalize the gradient to keep a constant speed even close to stationary points and flat regions. This is consistent with similar observations on normalized gradient methods (see Sec. III, [3]). In fact, Adam also has a similar normalization effect as it reduces to sign gradient descent when the momentum effect is turned off.
>
> Given this, one can replace sign with other types of function $\phi(x)$ that have similar normalization effect. In fact, if we view the sign as the derivative of $\ell_1$ norm, then we can generalize it to the derivation of a $\ell_p$ norm:
>
> $$\phi(x, p) = \big(\frac{|x|}{|x|_p}\big)^{p-1} \times \text{sign}(x) \times n^{\frac{p-1}{p}},$$
>
> where $n$ denotes the total number of parameters. Note that $p=1$ corresponds to LION and $p=2$ corresponds to LION-L2.
>
>
> [3] Revisiting Normalized Gradient Descent: Fast Evasion of Saddle Points.

---

### Official Review · Reviewer_fjaH · 2023-11-03

**Soundness:** 4 excellent
**Presentation:** 4 excellent
**Contribution:** 3 good
**Rating:** 8
**Confidence:** 3

**Summary:**

Authors use the analysis of the Lyapunov function to prove the theoretical guarantee for a broader family of Lion-K algorithms, in which the lion algorithm is part of. Authors also discuss connections of Lion-K with existing algorithms. They also emphasize it is still lack on the physical intuition on this algorithm.

**Strengths:**

This paper is very well written and super clear. Authors make every effort to have all the points stated out in a very concise way, even for the audience without too much background. The key contribution is they found the Lyapunov function for this family which is completely non-trivial. The guessing requires lots of trial and error. With the Lyapunov function, the optimization schemes are theoretically guaranteed

**Weaknesses:**

To demonstrate efficiency in the conclusion, authors didn't compare with other methods. At least, it is necessary to include other benchmark algorithms in empirical evaluations.

**Questions:**

1. Can you comment on the convergent rate of lion and lion-K algorithms, based on the current results? Why it performs comparable or favorably to AdamW?

2. Can you comment the possible convergence on lion-K vs interesting decomposition (12)? It seems they may both work well.

---

> ### Author Response · Authors · 2023-11-23
>
> We thank the reviewer for your helpful comments. We have conducted additional experiments and included these in the revised draft.
>
>  From Theorem 3.1, we have that (the rolling average of) $\Delta(x_t, m_t)$ converges with a $O(1/t)$. This is parallel to the standard results for classical gradient descent in which the gradient norm converges with $O(1/t)$. Different choices of $\mathcal{K}$ change the function $\Delta(x,m)$ (e.g., $\ell_1$ vs. $\ell_2$ norm), and hence are not completely comparable.
>
>    Meanwhile, we think that the current standard (worst-case) convergence rate is too rough and assumption-dependent to explain the practical power of optimization algorithms such as AdamW in deep learning. It is likely that better algorithms may only have a better constant, rather than a better convergence rate. Hence, we do not attempt to make arguments on convergence rate.
>
>    We think that having a Lyapunov function posits a necessary condition for an update rule to be a valid optimizer. It is then up to empirical results and insights to determine whether the algorithm performs well on the different problems of interest.
>
> We agree on the importance of comparing our method with benchmark algorithms. In the final manuscript, we will include a comparative analysis with established benchmarks to demonstrate the efficiency and effectiveness of our approach in a broader context.

---

### Official Review · Reviewer_GtGX · 2023-11-05

**Soundness:** 4 excellent
**Presentation:** 4 excellent
**Contribution:** 4 excellent
**Rating:** 8
**Confidence:** 2

**Summary:**

This paper studies the Lion optimizer, which is discovered through program search. It is shown that Lion indeed minimizes the loss while constraining the $\ell_\infty$ norm of the parameters. Specifically, the authors proposed a general framework for analyzing a general class of Lion-type algorithms, which lead to solutions of general composite optimization problems determined by reshaper function. Theoretical analyses revealed the two-phase dynamics of such algorithms: In the first phase, the iterates convergences to the constrained domain exponentially fast; then in the second phase, the dynamics minimizes the objective.

**Strengths:**

The paper is well written and very easy to follow. The proposed framework is very interesting, not only demestifying the Lion optimizer, but also encompassing many other algorithms. The authors have clearly explained the idea and discussed thoroughly the related background. The results seem solid and novel.

**Weaknesses:**

1. The current analyses apply to only algorithms with full gradient. It would be interesting to see results for Lion with stochastic gradients.
2. It would be helpful if the authors can discuss and comment on the implications of Theorem 4.1. Also, it is worth explaining why it is necessary to use a different implicit scheme.
3. Typo:
    - In the first sentence in the paragraph above Figure 4, "difference $\mathcal{K}$" -> "different $\mathcal{K}$"
    - The second paragraph above Lemma 2.1, "a systematic introduce to ..." -> " a systematic introduction to ..."

**Questions:**

In Section 3.1, the authors discuss the connection with existing algorithms. Do the corresponding convergence results also reduce to those classic ones?

---

> ### Author Response · Authors · 2023-11-23
>
> Dear reviewer, thank you very much for your comments. We have addressed the typos and further refined the draft.
>
> 1) The extension to the stochastic gradient case is straightforward given our results with the full gradient. We provide a new theorem in the Appendix (Check Theorem B.16 at page 30).
>
> 2) One convergence rate result implied from our framework is that $\Delta_1$ and $\Delta_2$ converge with an $O(1/t)$ rate. This is consistent with the results of classical algorithms on non-convex functions that the gradient (and momentum) norm converges to $O(1/t).$
>
>    It is also straightforward to obtain stronger convergence rates by placing stronger assumptions on $f$ (e.g., convexity or strong convexity), but since $f$ is non-convex in deep learning, we defer such derivations to future works.
>
> 3) We added comments on Theorem 4.1. The main implication is that we get parallel results on the two-phase convergence for the discrete-time case. The main difference is that the rolling average of $\Delta^1_t$ and $\Delta^2_t$ converges with $O(1/(\epsilon t) + \epsilon)$, where epsilon is the step size, rather than $O(1/t)$ in the continuous case.
>
>    The implicit form fits nicer with our proof as we tried to get a Lyapunov form in the discrete-time. We think it is mainly for technical reasons of the proof and as we explain in Section 4, the implicit form is equivalent to the explicit form with a different step size.

---

### Author Response · Authors · 2023-11-23
**General Response**

We sincerely appreciate all reviewers' valuable feedback and suggestions. We have revised the paper and the modified part is colored in green. Below, we summarize the additional experiments we conducted and address specific questions raised by reviewers.

### Additional Experiments:

**Experiment I (The Implication of Weight Decay):**
- **Section & Figure Reference:** Page 4-5, Figure 1 and Figure 4.
- **Summary:** We discuss the optimal weight decay $\lambda$ leading to fast convergence without sacrificing training loss. We demonstrate that a larger $\lambda$ generally accelerates learning, but performance drops when $\lambda$ exceeds a critical value $\lambda_0$. This finding underscores the importance of tuning $\lambda$ for Lion.

**Experiment II (Toy Example with Different $\mathcal{K}$):**
- **Section & Figure Reference:** Section 5, Page 10 -11, Figure 1 and Figure 5.
- **Summary:** We examine Lion's behavior with different $\mathcal{K}$ values in a similar toy example from Figure 1. This demonstrates that the observed behavior aligns well with our theoretical predictions.

**Experiment III (Lion-$\ell_p$ on ImageNet and Language Modeling):**
- **Section & Figure Reference:** Section 5, Figure 7.
- **Summary:** We evaluate Lion-$\ell_p$ with $p \in $ {1.0, 1.2, 1.4, 1.6, 1.8, 2.0} on ImageNet and Language Modeling tasks using ResNet-50, ViT-16B, and GPT-2. The results are intriguing: Lion-$\ell_p$ with smaller $p$ values performs better for ViT and GPT-2, but the opposite trend is observed with ResNet-50 on ImageNet. This suggests a potential architecture-dependence of the optimal $\mathcal{K}$, marking an interesting avenue for future research.

---

### Meta-Review · Area_Chair_AH2d · 2023-12-04

**Metareview:**

This paper aims at quantifying the convergence of a recently proposed optimizer, Lion. Both continuous- and discrete-time analyses are provided based on Lyapunov approach, and it is suggested that Lion implicitly poses an L-infinity constraint on the independent variable of the objective function. The dynamics is also shown to admit a Hamiltonian mirror descent structure.

Reviewers and I all agree that the paper is innovative, the theoretical contributions are solid, and the empirical demonstration is interesting. I'm thus delighted to recommend acceptance with some distinction. Nevertheless, I still recommend the authors take the reviewer-author discussion into consideration when preparing the camera-ready version.

**Justification For Why Not Higher Score:**

Several reviewers still seem to have some reservations.

**Justification For Why Not Lower Score:**

The paper is innovative, the theoretical contributions are solid, and the empirical demonstration is interesting.

---

### Decision · Program_Chairs · 2024-01-16

Accept (spotlight)